# On the Convergence of CART under Sufficient Impurity Decrease Condition

**Rahul Mazumder** [*]
Sloan School of Management
Massachusetts Institute of Technology
rahulmaz@mit.edu

**Haoyue Wang**
Operations Research Center
Massachusetts Institute of Technology
haoyuew@mit.edu

## Abstract

The decision tree is a flexible machine learning model that finds its success in numerous applications. It is usually fitted in a recursively greedy manner using CART. In this paper, we investigate the convergence rate of CART under a regression setting. First, we establish an upper bound on the prediction error of CART under a sufficient impurity decrease (SID) condition [10] – our result improves upon the known result by [10] under a similar assumption. Furthermore, we provide examples that demonstrate the error bound cannot be further improved by more than a constant or a logarithmic factor. Second, we introduce a set of easily verifiable sufficient conditions for the SID condition. Specifically, we demonstrate that the SID condition can be satisfied in the case of an additive model, provided that the component functions adhere to a "locally reverse Poincaré inequality". We discuss several well-known function classes in non-parametric estimation to illustrate the practical utility of this concept.

## 1 Introduction

The decision tree [5] is one of the most fundamental and popular methods in the machine learning toolbox. It utilizes a flowchart-like structure to recursively partition the data and allows users to derive interpretable decisions. It is usually constructed in a data-dependent greedy manner, with an algorithm called CART [5]. Thanks to its computational efficiency and ability to capture nonlinear structures, decision trees have served as the foundation for various influential algorithms for ensemble learning, including bagging [6], random forest [7] and gradient boosting [11]. These algorithms are among the best-known nonparametric models for supervised learning with tabular data [14].

Despite its remarkable empirical success in various applications, our theoretical understanding of decision trees remains somewhat limited. The data-dependent splits employed in CART pose challenges for rigorous theoretical analysis. While some easy-to-analyze variants of CART have been widely studied [18, 4, 3], rigorous theoretical analysis of the original version of CART has been absent until recently. In particular, a recent line of work [19, 24, 10, 16] has established the consistency and prediction error bounds for CART and random forest. See Section 1.1 for a comprehensive review of related literature.

In this paper, we study the decision tree under a nonparametric regression model:

$$y_i = f^*(x_i) + \epsilon_i \quad i \in [n] \tag{1}$$

with $x_i = (x_i^{(1)}, ..., x_i^{(p)}) \in \mathbb{R}^p$, $y_i \in \mathbb{R}$; $\epsilon_i$ is the noise with zero-mean, and $f^*(x)$ is the signal function. We consider a random design where $\{x_i\}_{i=1}^n$ are i.i.d. from some underlying distribution $\mu$ supported on $[0,1]^p$, and $\{\epsilon_i\}_{i=1}^n$ are i.i.d. independent of $\{x_i\}_{i=1}^n$. For an estimator $\hat{f}$ of $f^*$, we use

---

[*]This work is supported by Office of Naval Research (N00014-21-1-2841).

37th Conference on Neural Information Processing Systems (NeurIPS 2023).

the $L^2$ prediction error $\|\hat{f} - f^*\|_{L^2(\mu)}^2$ as a measure of the estimator $\hat{f}$. Consider a regression tree algorithm that produces a heuristic (not necessarily exact) solution $\hat{f}$ of

$$\min_{f \in \mathcal{T}_d} \quad \sum_{i=1}^{n} (f(x_i) - y_i)^2 \tag{2}$$

where $\mathcal{T}_d$ is the space of binary (i.e. each splitting node has two children) axis-aligned regression trees with maximum depth $d$. The statistical performance of the estimator $\hat{f}$, therefore, is affected by the following three factors: (i) Approximation error: how accurately can $f^*$ be approximated by a regression tree in $\mathcal{T}_d$. (ii) Estimation error: the variance of the estimator, which is affected by the complexity of the function space $\mathcal{T}_d$. A larger space $\mathcal{T}_d$ (i.e. larger $d$) can reduce the approximation error, but it may also increase the estimation error and lead to overfitting. (iii) Optimization error: the discrepancy between $\hat{f}$ and the exact optimal solution of the optimization problem (2). Indeed, if $\hat{f}$ is the exact optimal solution of (2) (i.e. optimal regression tree), it can approximate a broad class of $f^*$ [9], and the prediction error can be analyzed via the classical theory of least-squares estimators [22, 13]. However, since solving (2) to optimality is computationally challenging for large-scale data, heuristics like CART are typically employed in practice. To understand the statistical performance of CART, the major difficulty lies in the analysis of the optimization error of CART.

As a greedy algorithm, CART examines only one split in each iteration (see Section 1.3 for details). Although decision trees are intended to capture multivariate structures across different features, the greedy nature of CART poses challenges in approximating functions $f^*$ that have complex multivariate structures. For example, consider the two-dimensional "XOR gate" function $f^*(u) :=$ $1_{\{u \in [0,1/2)^2\}} + 1_{\{u \in [1/2,1)^2\}}$ [14], and assume that $x_i$ has a uniform distribution on $[0,1]^2$. Note that $f^*$ itself is a depth-2 decision tree, with (e.g.) the root node split with the first coordinate at $1/2$, and the left and right children split with the second coordinate at $1/2$. However, due to the symmetry structure of $f^*$, when the sample size $n$ is large, the root split of CART is completely fitting to noises and is likely to generate a split near the boundary [5, 8]. Consequently, the resulting estimator $\hat{f}$ fails to capture the true tree structure encoded in $f^*$. The major difficulty in fitting the XOR gate function with CART lies in the complicated interaction between the two coordinates.

The discussions above naturally lead to the following two questions about CART:

- What conditions on $f^*$ are needed such that CART can approximate $f^*$ and yield a consistent estimator?
- If $f^*$ satisfies such conditions, what is the convergence rate of the prediction error of CART?

In this paper, we aim to address these fundamental questions by providing sufficient conditions on $f^*$ such that one can establish convergence rates of CART. These sufficient conditions are quite flexible to include a broad class of nonparametric models. In the following, we first discuss some related literature and the basics of CART. In Section 2, we prove a general error bound for CART under a *sufficient impurity decrease (SID)* condition [10] of $f^*$. This error bound is an improvement over the existing results under similar conditions [10]. In Section 3, we introduce a sufficient condition for the SID condition on $f^*$, and use it to show that the SID condition can be satisfied by a broad class of $f^*$.

## 1.1 Related literature

The inception and analysis of decision trees and CART algorithm can be traced back to the seminal work of Breiman [5]. Breiman's study marked the first significant step towards establishing the consistency of CART, albeit under stringent assumptions concerning tree construction. The original formulation of CART, characterized by data-dependent splits, poses challenges when it comes to rigorous mathematical analysis. To address this issue, subsequent research efforts [18, 4, 3] have introduced alternative variants of CART that lend themselves to more tractable theoretical examinations. For instance, certain studies [12, 4, 3, 2] assume random selection of splitting features in the tree. Other approaches incorporate random selections of splitting thresholds [12, 2], while some restrict splits to occur exclusively at the midpoint of the corresponding cell in the chosen coordinate [3]. These simplifications ensure the independence between the splitting rules of the tree and the training data – the tree depends on the data only via the estimation in each cell. Such independence simplifies the analysis of the resultant trees. In a similar vein, alternative strategies have

been proposed, such as the employment of a two-sample approach [23] to compute splitting rules and cell estimations separately, giving rise to what is known as an "honest tree". This methodology also ensures independence between various aspects of the tree construction and estimation process.

Since the first consistency result in [5], the strict theoretical analysis of the original CART algorithm has remained elusive until recent advancements. A significant breakthrough came with the seminal work of Scornet et al. [19], which demonstrated the consistency of CART under the assumption of an additive model [15] with continuous component functions. It makes a technical contribution on the decrease of optimization error of CART. This technical argument strongly relies on the additive model. Subsequent works [17, 16] further extended the theoretical analysis by providing non-asymptotic error bounds for CART under a similar additive model. However, the rates derived in [17, 16] is a slow rate of $O((\log(n))^{-1})$, which seems not easy to be improved under the same assumption [8]. Although the additive model is a well-studied model in statistical learning, it is not immediately evident why such a modeling assumption is specifically required for CART to function effectively.

Another line of work [24, 10] adopts more intuitive assumptions on the signal function $f^*$ to ensure the progress of CART. In particular, Chi et al. [10] introduced a sufficient impurity decrease (SID) condition. In essence, the SID condition posits that for every rectangular region $B$ within the feature space, there exists an axis-aligned split capable of reducing the population variance of $f^*$ (conditional on $B$) by a fixed ratio. Note that such a split cannot be directly identified by CART, as it necessitates knowledge of the population distribution of $x_i$. Instead, CART employs empirical variance of $y_{i\,i=1}^n$ to guide its cell splitting decisions. Nonetheless, the SID condition is a strong assumption on the approximation power of tree splits, which can ensure the consistency of CART via an empirical process argument [10]. A similar condition has also appeared implicitly in [24]. Under the SID condition, Chi et al. [10] studied the consistency and convergence rates of CART, demonstrating a polynomial convergence $O(n^{-\gamma})$ of the prediction error. A possible limitation of the SID condition is the difficulty of verification: it necessitates the decreasing condition to hold true for all rectangles. Although a few examples are provided in [10], the general procedure for verifying whether a specific $f^*$ satisfies this condition remains unclear. Notably, its relationship with the additive model assumption in [19, 17] remains to be fully understood.

Lastly, it is worth mentioning that several recent studies [20, 8] have investigated the performance lower bounds of CART, shedding light on its limitations. Specifically, Tan et al. [20] demonstrated that even when assuming an additive regression function $f^*$, any single-tree estimator still suffers from the curse of dimensionality. Consequently, these estimators can only achieve convergence rates considerably slower than the oracle rates established for additive models [21]. Another study by Cattaneo et al. [8] delved into the splitting behavior of CART and provided theoretical support for empirical observations [5] indicating that CART tends to generate splits near boundaries in cases where the signal is weak. These lower bounds support the importance of using trees in ensemble models to reduce the prediction error. Although we focus on analyzing CART for a single tree in this paper, the result can be adapted to the analysis of ensembles of trees for the fitting of more complex function classes.

## 1.2 Contributions

The first contribution of our paper is a refined analysis of CART under the SID condition. Although the convergence of CART has been studied in [10] under SID condition, the analysis of [10] seems not tight when the noises have light-tails (see discussions in the Appendix C). We establish an improved analysis of CART under this setting, and show by examples that our error bound is tight up to log factors.

The second contribution of this paper is the decoding of the mystery of the SID condition. We discuss a sufficient condition under which the SID condition holds true. In particular, we introduce a class of univariate functions that satisfy an *locally reverse Poincaré inequality*, and show that additive functions with each univariate component being in this class satisfy the SID condition. This builds a connection between the two types of assumptions in the literature: additive model and SID condition. We discuss a few examples that how the locally reverse Poincaré inequality can be verified.

## 1.3 Basics of CART

We review the methodology of CART to build the tree. The primary objective of decision trees is to find optimal partitions of feature space that produce a minimal variance of response variables. CART constructs the tree and minimizes the variance using a top-down greedy approach. It starts with a root node that represents the whole feature space $\mathbb{R}^p$ – this can be viewed as an initial tree with depth 0. At each iteration, CART splits all the leave nodes in the current tree into a left child and a right child; the depth of the tree increases by 1, and the number of leave nodes in the tree doubles. At each leave node, it takes an *axis-aligned splits* with maximum variance (impurity) decrease. Let $\mathcal{E}$ be the set of all intervals (which can be open, closed, or open on one side and closed on another side). Define the set of rectangles in $[0,1]^p$ as:

$$\mathcal{A} := \Big\{ \prod_{j=1}^{p} E_j \ \Big| \ E_j \in \mathcal{E} \ \forall j \in [p] \Big\} \tag{3}$$

For each $A \in \mathcal{A}$, define $\mathcal{I}_A := \{i \in [n] \mid x_i \in A\}$. For a leave node that represents a rectangle $A \in \mathcal{A}$, the impurity decrease of a split in feature $j \in [p]$ with threshold $b \in \mathbb{R}$ is given by:

$$\widehat{\Delta}(A, j, b) := \frac{1}{n} \sum_{i \in \mathcal{I}_A} (y_i - \bar{y}_{\mathcal{I}_A})^2 - \frac{1}{n} \sum_{i \in \mathcal{I}_{A_L}} (y_i - \bar{y}_{\mathcal{I}_{A_L}})^2 - \frac{1}{n} \sum_{i \in \mathcal{I}_{A_R}} (y_i - \bar{y}_{\mathcal{I}_{A_R}})^2 \tag{4}$$

where $y_{\mathcal{I}} := \frac{1}{|\mathcal{I}|} \sum_{i \in \mathcal{I}} y_i$ for any $\mathcal{I} \subseteq [n]$, and

$$\begin{aligned} A_L = A_L(j, b) := A \cap \{v \in [0,1]^p \mid v_j \leq b\}, \\ A_R = A_R(j, b) := A \cap \{v \in [0,1]^p \mid v_j > b\}. \end{aligned} \tag{5}$$

To split on $A$, CART takes $\hat{j}$ and $\hat{b}$ that maximize the impurity decrease, i.e.,

$$\hat{j}, \hat{b} \in \operatorname*{argmax}_{j \in [p], b \in [0,1]} \widehat{\Delta}(A, j, b) \tag{6}$$

This splitting procedure is repeated until the tree has grown to a given maximum depth $d$.

## 1.4 Notations

For $a, b > 0$, we write $a = O(b)$ if there exists a universal constant $C > 0$ such that $a \leq Cb$; we write $b = \Omega(a)$ if $a = O(b)$, and write $a = \Theta(b)$ if both $a = O(b)$ and $a = \Omega(b)$ are true. For values $a_{n,p}, b_{n,p}$ that may depend on $n$ and $p$, we write $a_{n,p} = \widetilde{O}(b_{n,p})$ if $a_{n,p} = O(b_{n,p} \log^{\gamma}(np))$ for some fixed constant $\gamma \geq 0$.

## 2 Error bound of CART under SID property

We study the prediction error bound of CART for the regression problem (1). We focus on a random design as in the following assumption.

**Assumption 2.1** *(i) (Random design) Suppose $\{x_i\}_{i=1}^n$ are i.i.d. random variables with a distribution $\mu$ supported on $[0,1]^p$. Suppose $\mu$ has a density $p_X$ (w.r.t. Lebesgue measure) on $[0,1]^p$ satisfying $0 < \underline{\theta} \leq p_X(u) \leq \bar{\theta} < \infty$ for all $u \in [0,1]^p$ for some constants $\underline{\theta} \leq 1$ and $\bar{\theta} \geq 1$.*

*(ii) (Error distribution) Suppose $\{\epsilon_i\}_{i=1}^n$ are i.i.d. zero-mean bounded random variables with $|\epsilon_i| \leq m < \infty$ for some $m > 0$. Suppose $\{\epsilon_i\}_{i=1}^n$ are independent to $\{x_i\}_{i=1}^n$.*

*(iii) (Bounded signal function) Suppose $\sup_{u \in [0,1]^p} |f^*(u)| \leq M < \infty$ for some constant $M$.*

In Assumption 2.1 (i), it is assumed that the features $x_i$ are supported on $[0,1]^p$ for convenience. The same results (different by at most a constant) can be proved if we replace $[0,1]^p$ with an arbitrary bounded rectangle. The second assumption (Assumption 2.1 (ii)) requires the error terms $\{\epsilon_i\}_{i=1}^n$ to be i.i.d. bounded – this is the key difference of our paper and the assumptions made in [10]. In particular, [10] made a milder assumption on the error terms that allows them to have heavy tails. Their result, however, when specifying to the case of bounded noises, is not tight (see

discussions in Appendix C for a comparison). Note that for convenience we have assumed that $\epsilon_i$ is bounded. If instead, we assume the noises are sub-Gaussian, similar conclusions can also be proved via a truncation argument (the bound may be different by a log factor). The third assumption (Assumption 2.1 (iii)) is a standard assumption for nonparametric regression models [22, 13].

Under the random design setting in Assumption 2.1, we are ready to introduce the sufficient impurity decrease condition on the underlying function $f^*$. Let $X$ be a random variable in $\mathbb{R}^p$ that has the same distribution as $x_1$ and is independent to $\{x_1, ..., x_n\}$. For any $A \in \mathcal{A}$, $j \in [p]$ and $b \in \mathbb{R}$, define:

$$\begin{aligned}\Delta(A, j, b) := \mathbb{P}(X \in A)\text{Var}(f^*(X)|X \in A) &- \mathbb{P}(X \in A_L)\text{Var}(f^*(X)|X \in A_L) \\ &- \mathbb{P}(X \in A_R)\text{Var}(f^*(X)|X \in A_R)\end{aligned} \tag{7}$$

where $A_L$ and $A_R$ are defined in (5), and $\text{Var}(f^*(X)|X \in A)$ means the conditional variance defined as

$$\text{Var}(f^*(X)|X \in A) := \mathbb{E}\Big(\big(f^*(X) - \mathbb{E}(f^*(X)|X \in A)\big)^2\Big|X \in A\Big) \tag{8}$$

It is not hard to see the similarity between $\Delta(A, j, b)$ and $\widehat{\Delta}(A, j, b)$ (defined in (4)), where the former can be viewed as a population variant of the latter. Intuitively, for a split with feature $j$ and threshold $b$, $\Delta(A, j, b)$ measures variance decrease of $f^*$ after this split. The SID condition below assumes that there is a split with a large variance decrease.

**Assumption 2.2** *(Sufficient Impurity Decrease) There exists a constant $\lambda \in (0, 1]$ such that for all $A \in \mathcal{A}$,*

$$\sup_{j\in[p],b\in\mathbb{R}} \Delta(A, j, b) \geq \lambda \cdot \mathbb{P}(X \in A)\text{Var}(f^*(X)|X \in A) \tag{9}$$

Note that Assumption 2.2 is a condition depending on both the function $f^*$ and the underlying distribution $X \sim \mu$. Briefly speaking, the SID condition requires that the best split must decrease the population variance by a constant factor (note that $\lambda$ is required to be bounded away from 0). Intuitively, the condition (9) can be satisfied if $f^*$ has a significant "trend" along some axis in $A$, but may be hard to be satisfied if $f^*$ is relatively "flat" in $A$. Since Assumption 2.2 requires (9) being satisfied for all cells $A \in \mathcal{A}$, it essentially requires that $f^*$ is not "too flat" on any local rectangle $A$. In Section 3, we develop a technical argument to translate this intuition into a rigorous statement, and use it to check Assumption 2.2 for a wide class of functions.

With Assumptions 2.1 and 2.2 at hand, we are ready to present the main result in this section. For any function $g : \mathbb{R}^p \to \mathbb{R}$, let $\|g\|_{L^2(\mu)}$ denote the $L^2$-norm of $g$ with respect to the measure $\mu$.

**Theorem 2.3** *Suppose Assumptions 2.1 and 2.2 hold true. Let $\widehat{f}^{(d)}(\cdot)$ be the tree estimated by CART with depth $d$. Suppose $n$ is large enough such that $\frac{2\bar{\theta}e^2d}{n} \vee \frac{\log(72p^d(n+1)^{2d}/\delta)}{n} < 3/4$. Then there exists a universal constant $C > 0$ such that with probability at least $1 - \delta$, it holds that for any $\alpha > 0$,*

$$\|\widehat{f}^{(d)} - f^*\|_{L^2(X)}^2 \leq 2\text{Var}(f^*(X)) \cdot \Big(1 - \frac{\lambda}{(1+\alpha)^2}\Big)^d + C\frac{2^d(d\log(np) + \log(1/\delta))}{\alpha n}U^2 \tag{10}$$

*where $U := M + m$. In particular, taking $\alpha = 1/d$ and $d = \lceil \log_2(n)/(1 - \log_2(1-\lambda)) \rceil$, it holds*

$$\|\widehat{f}^{(d)} - f^*\|_{L^2(X)}^2 \leq C_{\lambda,U} \frac{\log^2(n)\log(np) + \log(n)\log(1/\delta)}{n^{\phi(\lambda)}} \tag{11}$$

*where $\phi(\lambda) := \frac{-\log_2(1-\lambda)}{1-\log_2(1-\lambda)}$, and $C_{\lambda,U}$ is a constant that only depends on $\lambda$ and $U$.*

The error bound in (10) consists of two terms. The first term corresponds to the bias of the approximating $f^*$ using CART. It decreases geometrically as $d$ increases, which is suggested by the SID condition. The second term is $O(2^d d\log(np)/n)$, which corresponds to the estimation variance of CART. Here the term $2^d$ represents the model complexity – a fully-grown depth $d$ tree has $2^d$ leaves. The proof of Theorem 2.3 is presented in Appendix A. It is based on a careful technical argument that controls the difference between the population impurity decrease $\Delta(A, j, b)$ and the empirical impurity decrease $\widehat{\Delta}(A, j, b)$. In particular, our analysis is different from that of [10]. Note that [10] makes a different assumption on the error distribution, and their technical argument is not sufficient for the proof of Theorem 2.3. Particularly, the error bound in (11) is an improvement of the results proved in Theorem 1 of [10]—see Section C for a detailed comparison.

Note that the convergence rate in (11) has a crucial dependence on the coefficient $\lambda$. For a large value of $\lambda$, the exponent $\phi(\lambda)$ is also larger, leading to a faster convergence rate in (11). On the other end, if $\lambda$ is small, as $\lambda \to 0$, we have $\phi(\lambda) \to 0$, and the convergence rate in (11) can be arbitrarily slow. Nonetheless, as long as the SID condition is satisfied with $\lambda$ bounded away from $0$, (11) shows a polynomial rate, which is better than the logarithmic rate in [17, 16] without the SID condition.

To explore the tightness of the error bounds in Theorem 2.3, we consider a simple example with $f^*$ being a univariate linear function.

**Example 2.1** *(Univariate linear function) Suppose $p = 1$, and $x_i$ are i.i.d. from the uniform distribution on $[0, 1]$. Suppose $f^*$ is a univariate linear function: $f^*(t) = t$. Then the SID condition (Assumption 2.2) is satisfied with coefficient $\lambda = 3/4$. As a result, it holds $\phi(\lambda) = 2/3$, and the error bound in the RHS of (11) is $\widetilde{O}(n^{-2/3})$.*

*Proof.* For any interval $A = [\ell, r]$, we have

$$\mathbb{P}(X \in A)\mathrm{Var}(f^*(X)|X \in A) = \int_\ell^r \left(t - \frac{\ell + r}{2}\right)^2 \mathrm{d}t = \frac{(r - \ell)^3}{12} \tag{12}$$

Take $b := (\ell + r)/2$, then we have

$$\begin{aligned}
\Delta(A, 1, b) &= \int_\ell^r \left(t - \frac{\ell + r}{2}\right)^2 \mathrm{d}t - \int_\ell^b \left(t - \frac{\ell + b}{2}\right)^2 \mathrm{d}t - \int_b^r \left(t - \frac{b + r}{2}\right)^2 \mathrm{d}t \\
&= \frac{1}{12}(r - \ell)^3 - \frac{1}{12}(b - \ell)^3 - \frac{1}{12}(r - b)^3 = \frac{3}{48}(r - \ell)^3
\end{aligned} \tag{13}$$

Combining (12) and (13) we know that Assumption 2.2 is satisfied with $\lambda = 3/4$. $\square$

As shown by Example 2.1, when $f^*$ is a univariate linear function and $\{x_i\}_{i=1}^n$ are from a uniform distribution, the error bound in (11) reduces to $\widetilde{O}(n^{-2/3})$. This rate matches the lower bound for any partition-based estimator (see e.g. [13] Chapter 4), [2] hence showing that the upper bound in (11) cannot be improved (by more than a log factor) via any refined analysis. To verify the rate $\widetilde{O}(n^{-2/3})$, we conduct a simulation shown by Figure 1. In particular, Figure 1 presents the log-log plot of the prediction error of CART versus different sample sizes, where the depth $d$ is taken as stated above (11). A reference line of slope $-2/3$ is also shown. It can be seen that the slope of the line for CART is also roughly $-2/3$, which is consistent with the rate $\widetilde{O}(n^{-2/3})$ derived in Example 2.1.

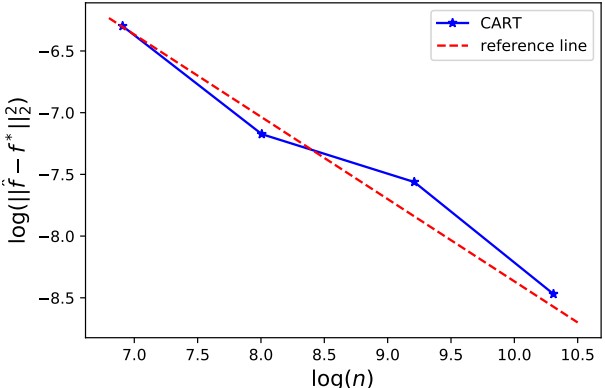

Figure 1: Prediction error of CART versus sample size $n$ for a univariate linear model.

---

[2]There is a subtle difference between the lower bound in [13] Chapter 4 and the upper bound in Theorem 2.3. The lower bound in [13] requires that the splitting points of the partition are independent of the training data, while for CART this condition is not satisfied.

# 3 Function classes satisfying SID condition

In this section, we introduce a class of sufficient conditions that facilitate the easy verification of the SID condition for a broad range of functions. Specifically, we focus on the case when $f^*$ has an additive structure:

$$f^*(u) = f_1^*(u_1) + f_2^*(u_2) + \cdots + f_p^*(u_p) \tag{14}$$

where $f_j^*$'s are univariate functions. Note that additive model has also been assumed in prior works such as [19, 17, 16] for the study of CART. If we only assume that each component function $f_j^*$ has bounded total variation, then only a slow rate $O(\log^{-1}(n))$ is known [17]. In the following, we discuss a few sufficient conditions concerning the component functions $f_j^*$ under which $f^*$ satisfies the SID condition, consequently guaranteeing a faster convergence rate as indicated by Theorem 2.3.

First, we introduce a key class of univariate functions that satisfy a special integral inequality.

**Definition 3.1** *(Locally Reverse Poincaré Class) For a univariate differentiable function $g$ on an interval $Q \subset \mathbb{R}$, we say it belongs to the Locally Reverse Poincaré (LRP) class on $Q$ with parameter $\tau$ if for any subinterval $[a, b] \subseteq Q$,*

$$\left( \int_a^b |g'(t)| \, \mathrm{d}t \right)^2 \leq \frac{\tau^2}{b-a} \inf_{w \in \mathbb{R}} \int_a^b |g(t) - w|^2 \, \mathrm{d}t \tag{15}$$

*We use the notation $LRP(Q, \tau)$ to denote the class of all such functions.*

It is worth noting that for any given univariate differentiable function $g$ and a fixed interval $[a, b]$, there exists a constant $\tau$ such that the reverse of inequality (15) holds true, as established by the Poincaré inequality. The condition (15) requires a uniform constant $\tau > 0$ such that the reverse of the Poincaré inequality holds true. Note that the LHS of (15) is equivalent to the square of the total variation of $g$ on $[a, b]$, and the RHS is a constant multiple of the conditional variance of $g$ on $[a, b]$ (under the uniform distribution). Intuitively, this condition can be satisfied when the function $g$ exhibits a clear "trend" within any interval $[a, b]$ rather than being fuzzy fluctuations. We give a few examples below.

**Example 3.1** *(Strongly increasing function) Let $g$ be a strictly increasing function on $[0, 1]$ with $c_2 \geq g'(t) \geq c_1 > 0$ for all $t \in [0, 1]$. Then $g \in LRP([0, 1], 2\sqrt{3}c_2/c_1)$.*

Example 3.1 provides a direct generalization of the simple linear function showcased in Example 2.1. Note that this example was also discussed in [10]. In Example 3.1, since $g'(t)$ is consistently bounded away from zero, verifying the inequality 15 becomes straightforward. Below we present a more nuanced example where $g'(t)$ may equal zero at certain point $t \in [0, 1]$.

**Example 3.2** *(Smooth and strongly convex function) Let $g$ be a $L$-smooth and $\sigma$-strongly-convex function on $[0, 1]$ for some $L > \sigma > 0$, i.e.,*

$$\frac{\sigma}{2}(t-s)^2 \leq g(t) - g(s) - g'(s)(t-s) \leq \frac{L}{2}(t-s)^2 \tag{16}$$

*for all $t, s \in [0, 1]$. Then $g \in LRP([0, 1], 110(L/\sigma))$.*

Note that for a smooth and strongly convex function $g$, the gradient $g'(t)$ can be zero at some $t \in [0, 1]$. But the strong convexity condition guarantees that there is at most one point $t$ where $g'(t) = 0$, and the inequality (15) can still be satisfied. Additional details can be found in Appendix B.

**Example 3.3** *(Polynomial with degree $r$) There exists a constant $C_r$ that only depends on $r$ such that for any univariate polynomial $g$ of degree at most $r$, $g \in LRP(\mathbb{R}, C_r)$.*

Although the set of polynomials (with a degree at most $r$) appears to be complicated, it is a finite-dimensional linear space. Therefore the differentiation is a bounded linear operator between finite dimensional normed linear spaces. As a result, the conclusion of Example 3.3 can be proved via a variable transformation argument. See Appendix B for details.

Examples 3.1 – 3.3 demonstrate that the LRP condition can be readily verified for various well-known function classes, but the relationship between the LRP class and the SID condition remains unclear. Below we present the first main result in this section, which establishes a connection between the LRP class and the SID condition under the additive model.

**Proposition 3.1** *Suppose Assumption 2.1 holds true, and $f^*$ has an additive structure as in* (14). *If $f_k^* \in LRP([0,1], \tau)$ for all $k \in [p]$, then Assumption 2.2 is satisfied with $\lambda = 4\underline{\theta}/(p\tau^2\bar{\theta})$.*

As shown by Proposition 3.1, for the additive model (14), when the basic conditions in Assumption 2.1 are satisfied, the SID condition holds true for $f^*$ as long as each component function is in an LRP class. Particularly, the SID condition is satisfied for component functions from Examples 3.1 – 3.3. Note that the coefficient $\lambda$ in Proposition 3.1 is proportional to $1/p$, which implies that it can be very small when $p$ is large. In particular, with this $\lambda = \Theta(1/p)$ (with $\underline{\theta}, \bar{\theta}$ and $\tau$ being constants), by Theorem 2.3, the prediction error bound in the RHS of (11) is $O(n^{-c/p})$ for some constant $c > 0$ [3]. Since the exponent is dependent on $p$, the curse of dimensionality becomes evident. This seems to indicate that the analysis of Proposition 3.1 could potentially be refined. However, given the negative results demonstrated by [20, 8], it is likely that for tree-based estimators, even with the additive model, the curse of dimensionality cannot be circumvented.

Proposition 3.1 requires each component of the additive function to lie in an LRP class. Actually, this condition can be further relaxed as in the following proposition.

**Proposition 3.2** *Suppose Assumption 2.1 holds true, and $f^*$ has an additive structure as in* (14). *Given integer $r \geq 1$ and constants $\alpha > 0, \beta \geq 1$. Suppose for each $k \in [p]$ there exist $0 = t_0^{(k)} < t_1^{(k)} < \cdots < t_{r-1}^{(k)} < t_r^{(k)} = 1$ with $t_j^{(k)} - t_{j-1}^{(k)} \geq \alpha/r$ and $f_k^* \in LRP([t_{j-1}, t_j], \beta)$ for all $j \in [r]$. Then Assumption 2.2 is satisfied with parameter $\lambda = \underline{\theta}/(p\bar{\theta}\tau^2)$ where $\tau^2 = \max\left\{\frac{2r\bar{\theta}}{\underline{\theta}}, \frac{r^2}{2\alpha}\right\}\max\{9\beta^2, 32 + \beta^2\}$.*

By Proposition 3.2, an additive function $f^*$ satisfies the SID condition if each component function is a piecewise function, with each piece belonging to an LRP class. Notably, continuity at the joining points of consecutive pieces is not necessary. However, for technical reasons, it is required that each piece has a length that is not excessively small, as determined by the parameter $\alpha$. See Appendix B for the proofs of Proposition 3.1 and 3.2.

## 4 Proof sketch of Theorem 2.3

We provide a sketch of the proof of Theorem 2.3. For $p \geq 1$ and $d \geq 1$, define

$$\mathcal{A}_{p,d} := \left\{ \prod_{j=1}^{p} [\ell_j, u_j] \in \mathcal{A} \ \Big| \ \#\{j \in [p] \mid [\ell_j, u_j] \neq [0,1]\} \leq d \right\} \tag{17}$$

That is, each rectangle in $\mathcal{A}_{p,d}$ has at most $d$ dimensions that are not the full interval $[0, 1]$. When $p \geq d$, $\mathcal{A}_{p,d}$ contains all the rectangles in $[0,1]^p$, i.e. $\mathcal{A}_{p,d} = \mathcal{A}$. However, when $d$ is much smaller than $p$ (in a high-dimensional setting), $\mathcal{A}_{p,d}$ contains much fewer rectangles than $\mathcal{A}$. Note that for a decision tree with depth $d$, each leaf node represents a rectangle in $\mathcal{A}_{p,d}$.

The proof of Theorem 2.3 builds upon a few technical lemmas which provide uniform bounds between the empirical and populational quantities. These results may hold their own significance in the study of other partition-based algorithms. For $\delta \in (0, 1)$, define values

$$\bar{t}_1(\delta) = \bar{t}_1(\delta, n, d) := \frac{4}{n}\log(2p^d(n+1)^{2d}/\delta)$$

$$\bar{t}_2(\delta) = \bar{t}_2(\delta, n, d) := \frac{2\bar{\theta}e^2 d}{n} \vee \frac{\log(p^d(n+1)^{2d}/\delta)}{n} \tag{18}$$

$$\bar{t}(\delta) = \bar{t}(\delta, n, d) := \bar{t}_1(\delta, n, d) \vee \bar{t}_2(\delta, n, d)$$

where $\bar{\theta}$ is the constant in Assumption 2.1 $(i)$. Note that we have $\bar{t}(\delta) \leq O\left(\frac{d\log(np)+\log(1/\delta)}{n}\right)$. We have the following uniform bounds on empirical and populational mean on every rectangle in $\mathcal{A}_{p,d}$.

**Lemma 4.1** *Suppose Assumption 2.1 holds true. Suppose $\bar{t}_2(\delta/12) < 3/4$. Then with probability at least $1 - \delta$, it holds*

$$\sup_{A \in \mathcal{A}_{p,d}} \sqrt{\mathbb{P}(X \in A)}\Big|\mathbb{E}(f^*(X)|X \in A) - \bar{y}_{\mathcal{I}_A}\Big| \leq 20U\sqrt{\bar{t}(\delta/12)} \tag{19}$$

---

[3]The lower bound is based on general additive models which may not satisfy the SID condition. When the SID condition is satisfied, the lower bound may be better.

For a tree-based estimator in practice, the prediction at each leaf node is usually given by the sample average of $y_i$ for all the samples $i$ routed to that leaf, i.e., the value $\bar{y}_A$ for a leaf with rectangle $A$. Therefore, Lemma 4.1 provides a uniform bound on the error if we replace the prediction at each leaf by the populational conditional mean $\mathbb{E}(f^*(X)|X \in A)$. It is worth noting that the gap between $\bar{y}_A$ and $\mathbb{E}(f^*(X)|X \in A)$ is rescaled by the quantity $\sqrt{\mathbb{P}(X \in A)}$. Intuitively, when the rectangle $A$ is small and hence $\mathbb{P}(X \in A)$ is small, few points are routed to the rectangle $A$. As a result, the corresponding gap between $\bar{y}_A$ and $\mathbb{E}(f^*(X)|X \in A)$ can be large – which leads to a large discrepancy of the estimate and the signal locally in this cell $A$. However, in another perspective, since the cell $A$ is small, a large discrepancy in $A$ only makes a small contribution to the overall error $\|\widehat{f}^{(d)} - f^*\|^2_{L^2(X)}$. Therefore, the factor $\sqrt{\mathbb{P}(X \in A)}$ in the LHS of (19) is a proper balance of these two aspects. This is the key technical argument that differs from the analysis in [10]. It helps establish the tighter bounds in Theorem 2.3.

**Lemma 4.2** *Suppose Assumption 2.1 holds true. Given a constant $\alpha > 0$. Given any $\delta \in (0,1)$, suppose $\bar{t}_2(\delta/36) < 3/4$. Then with probability at least $1 - \delta$, it holds*

$$\Delta(A,j,b) \le (1+\alpha)\widehat{\Delta}(A,j,b) + (1+1/\alpha) \cdot C'U^2\bar{t}(\delta/36) \quad \forall\, A \in \mathcal{A}_{p,d-1},\ j \in [p],\ b \in \mathbb{R}$$

$$\widehat{\Delta}(A,j,b) \le (1+\alpha)\Delta(A,j,b) + (1+1/\alpha) \cdot C'U^2\bar{t}(\delta/36) \quad \forall\, A \in \mathcal{A}_{p,d-1},\ j \in [p],\ b \in \mathbb{R}$$

*for some universal constant $C' > 0$.*

Lemma 4.2 establishes upper bounds on the discrepancy between empirical and populational impurity decrease. It builds an avenue to translate the SID condition into an empirical counterpart, which is further used to derive the decrease of objective value.

With Lemmas 4.1 and 4.2 at hand, we are ready to wrap up the proof of Theorem 2.3. First, we have

$$\|\widehat{f}^{(k)} - f^*\|^2_{L^2(X)} \le 2\|f^* - \widetilde{f}^{(k)}\|^2_{L^2(X)} + 2\|\widetilde{f}^{(k)} - \widehat{f}^{(k)}\|^2_{L^2(X)} := 2J_1(k) + 2J_2(k) \quad (20)$$

where $\widetilde{f}^{(k)}$ is a tree with the same splitting rule as $\widehat{f}^{(k)}$ but leaf predictions replaced by the populational means (in that leaf). The term $J_2(k)$ can be bounded as

$$J_2(k) = \sum_{t \in \mathcal{L}^{(k)}} \mathbb{P}(X \in A_t)\big(\mathbb{E}(f^*(X)|X \in A_t, \mathcal{X}_1^n) - \bar{y}_{\mathcal{I}_{A_t}}\big)^2 \le CU^2 2^k \cdot \frac{k\log(np) + \log(1/\delta)}{n}$$

where $\mathcal{L}^{(k)}$ is the set of leaves of $\widehat{f}^{(k)}$ at depth $k$; $A_t$ is the corresponding rectangle for a leaf $t$; and $C$ is a universal constant. The last inequality in (21) is by Lemma 4.1. The term $J_1(k)$ can be written as

$$J_1(k) = \sum_{t \in \mathcal{L}^{(k)}} \mathbb{P}(X \in A_t|\mathcal{X}_1^n) \cdot \mathrm{Var}(f^*(X)|X \in A_t, \mathcal{X}_1^n)$$

Making use of Lemma 4.2 and by some algebra (see Appendix A for details), a recursive inequality can be established:

$$J_1(k+1) \le \left(1 - \frac{\lambda}{(1+\alpha)^2}\right)J_1(k) + 2^k \cdot \frac{CU^2(k\log(np) + \log(1/\delta))}{n}$$

The proof of Theorem 2.3 is complete by telescoping the inequality above, and combining the upper bounds for $J_1(k)$ and $J_2(k)$ with (20).

## 5 Conclusion and discussions

We have proved an error bound on the prediction error of CART for a regression problem when $f^*$ satisfies the SID condition. This error bound cannot be improved by more than a log factor. We have also discussed a few sufficient conditions under which we can show that an additive model satisfies the SID condition.

One possible limitation of this work is that: it seems that the SID coefficients $\lambda$ derived in Section 3 are not tight. Since $\lambda$ appeared in the exponent of the rate in Theorem 2.3, a tighter estimate of $\lambda$ leads to an improved convergence rate in (11). We leave it as a future work for a more precise analysis of the SID condition.

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

# A Proof of Theorem 2.3

## A.1 Notations

We first define some notations in the context of the model (1). For $p \geq 1$ and $d \geq 1$, define

$$\mathcal{A}_{p,d} := \left\{ \prod_{j=1}^{p} [\ell_j, u_j] \in \mathcal{A} \;\middle|\; \#\{j \in [p] \mid [\ell_j, u_j] \neq [0,1]\} \leq d \right\} \tag{21}$$

That is, each rectangle in $\mathcal{A}_{p,d}$ has at most $d$ dimensions that are not the full interval $[0,1]$. Note that for a decision tree with depth $d$, each leave node represents a rectangle in $\mathcal{A}_{p,d}$. Furthermore, for $\delta \in (0,1)$, define values

$$\bar{t}_1(\delta) = \bar{t}_1(\delta, n, d) := \frac{4}{n} \log(2p^d(n+1)^{2d}/\delta)$$

$$\bar{t}_2(\delta) = \bar{t}_2(\delta, n, d) := \frac{2\bar{\theta}e^2 d}{n} \vee \frac{\log(p^d(n+1)^{2d}/\delta)}{n} \tag{22}$$

$$\bar{t}(\delta) = \bar{t}(\delta, n, d) := \bar{t}_1(\delta, n, d) \vee \bar{t}_2(\delta, n, d)$$

where $\bar{\theta}$ is the constant in Assumption 2.1 $(i)$. Note that we have $\bar{t}(\delta) \leq O(d \log(np/\delta)/n)$.

For two values $a, b > 0$, we write $a \lesssim b$ if there is a universal constant $C > 0$ such that $a \leq Cb$. We write $a \lesssim_r b$ if there is a constant $C_r$ that only depends on $r$ such that $a \leq C_r b$.

## A.2 Technical lemmas

Now we can introduce the major technical results to establish the error bound.

**Lemma A.1** *Suppose Assumption 2.1 holds true. Suppose $\bar{t}_2(\delta/12) < 3/4$. Then with probability at least $1 - \delta$, it holds*

$$\sup_{A \in \mathcal{A}_{p,d}} \sqrt{\mathbb{P}(X \in A)} \left| \mathbb{E}(f^*(X)|X \in A) - \bar{y}_{\mathcal{I}_A} \right| \leq 20U \sqrt{\bar{t}(\delta/12)} \tag{23}$$

The proof of Lemma A.1 is presented in Section A.4. Note that Lemma A.1 provides a uniform bound on the gap between the populational mean $\mathbb{E}(f^*(X)|X \in A)$ and the sample mean $\bar{y}_{\mathcal{I}_A}$. This is used to derive the geometric decrease of the bias, using the SID assumption.

**Lemma A.2** *Suppose Assumption 2.1 holds true. Given any $\delta \in (0,1)$, suppose $\bar{t}_2(\delta/4) < 3/4$. Then with probability at least $1 - \delta$ it holds*

$$\sup_{A \in \mathcal{A}_{p,d}} \left| \sqrt{\mathbb{P}(X \in A)} - \sqrt{|\mathcal{I}_A|/n} \right| \leq 5\sqrt{\bar{t}(\delta/4)} \tag{24}$$

The proof of Lemma A.2 is presented in Section A.5. Lemma A.2 provides a uniform deviation gap between the square root of probability and sample frequency over all sets in $\mathcal{A}_{p,d}$. Note that this uniform bound is stronger than a result without a square root (which can be obtained easily via Hoeffding's inequality and a union bound), and is useful to prove the final error bound in Theorem 2.3.

For any rectangle $A \in \mathcal{A}$, $j \in [p]$ and $b \in \mathbb{R}$, define

$$\Delta_L(A, j, b) := \mathbb{P}(X \in A_L)\Big(\mathbb{E}(f^*(X)|X \in A) - \mathbb{E}(f^*(X)|X \in A_L)\Big)^2$$

$$\Delta_R(A, j, b) := \mathbb{P}(X \in A_R)\Big(\mathbb{E}(f^*(X)|X \in A) - \mathbb{E}(f^*(X)|X \in A_R)\Big)^2$$

$$\widehat{\Delta}_L(A, j, b) := \frac{|\mathcal{I}_{A_L}|}{n}(\bar{y}_{\mathcal{I}_{A_L}} - \bar{y}_{\mathcal{I}_A})^2$$

$$\widehat{\Delta}_R(A, j, b) := \frac{|\mathcal{I}_{A_R}|}{n}(\bar{y}_{\mathcal{I}_{A_R}} - \bar{y}_{\mathcal{I}_A})^2$$

We have the following identity regarding the impurity decrease of each split.

**Lemma A.3** *For any rectangle $A \in \mathcal{A}$, $j \in [p]$ and $b \in \mathbb{R}$, it holds*

$$\Delta(A, j, b) = \Delta_L(A, j, b) + \Delta_R(A, j, b)$$

$$\widehat{\Delta}(A, j, b) = \widehat{\Delta}_L(A, j, b) + \widehat{\Delta}_R(A, j, b) \tag{25}$$

*Proof.* We just present the proof of the second equality. The proof of the first equality can be proved similarly. Note that

$$
\begin{aligned}
\widehat{\Delta}(A, j, b) &= \frac{1}{n} \sum_{i \in \mathcal{I}_A} (y_i - \bar{y}_{\mathcal{I}_A})^2 - \frac{1}{n} \sum_{i \in \mathcal{I}_{A_L}} (y_i - \bar{y}_{\mathcal{I}_{A_L}})^2 - \frac{1}{n} \sum_{i \in \mathcal{I}_{A_R}} (y_i - \bar{y}_{\mathcal{I}_{A_R}})^2 \\
&= \frac{1}{n} \sum_{i \in \mathcal{I}_{A_L}} \left[ (y_i - \bar{y}_{\mathcal{I}_A})^2 - (y_i - \bar{y}_{\mathcal{I}_{A_L}})^2 \right] + \frac{1}{n} \sum_{i \in \mathcal{I}_{A_R}} \left[ (y_i - \bar{y}_{\mathcal{I}_A})^2 - (y_i - \bar{y}_{\mathcal{I}_{A_R}})^2 \right]
\end{aligned}
\tag{26}
$$

For the first term, we have

$$
\begin{aligned}
&\frac{1}{n} \sum_{i \in \mathcal{I}_{A_L}} \left[ (y_i - \bar{y}_{\mathcal{I}_A})^2 - (y_i - \bar{y}_{\mathcal{I}_{A_L}})^2 \right] \\
&= \frac{1}{n} \sum_{i \in \mathcal{I}_{A_L}} \left[ (y_i - \bar{y}_{\mathcal{I}_A})^2 - (y_i - \bar{y}_{\mathcal{I}_A})^2 - 2(y_i - \bar{y}_{\mathcal{I}_A})(\bar{y}_{\mathcal{I}_A} - \bar{y}_{\mathcal{I}_{A_L}}) - (\bar{y}_{\mathcal{I}_A} - \bar{y}_{\mathcal{I}_{A_L}})^2 \right] \\
&= \frac{|\mathcal{I}_{A_L}|}{n} (\bar{y}_{\mathcal{I}_A} - \bar{y}_{\mathcal{I}_{A_L}})^2 = \widehat{\Delta}_L(A, j, b)
\end{aligned}
\tag{27}
$$

Similarly, we have

$$
\frac{1}{n} \sum_{i \in \mathcal{I}_{A_R}} \left[ (y_i - \bar{y}_{\mathcal{I}_A})^2 - (y_i - \bar{y}_{\mathcal{I}_{A_R}})^2 \right] = \widehat{\Delta}_R(A, j, b)
\tag{28}
$$

The proof is complete by combining (26), (27) and (28). $\qquad\square$

**Lemma A.4** *Suppose Assumption 2.1 holds true. Given a constant $\alpha > 0$. Given any $\delta \in (0, 1)$, suppose $\bar{t}_2(\delta/36) < 3/4$. Then with probability at least $1 - \delta$, it holds*

$$
\Delta(A, j, b) \le (1 + \alpha)\widehat{\Delta}(A, j, b) + (1 + 1/\alpha) \cdot 5000 U^2 \bar{t}(\delta/36) \quad \forall A \in \mathcal{A}_{p, d-1}, \ j \in [p], \ b \in \mathbb{R}
\tag{29}
$$

*and*

$$
\widehat{\Delta}(A, j, b) \le (1 + \alpha)\Delta(A, j, b) + (1 + 1/\alpha) \cdot 5000 U^2 \bar{t}(\delta/36) \quad \forall A \in \mathcal{A}_{p, d-1}, \ j \in [p], \ b \in \mathbb{R}
\tag{30}
$$

*Proof.* For $A \in \mathcal{A}_{p, d-1}, j \in [p]$ and $a \in \mathbb{R}$, by Lemma A.3 we have

$$
\begin{aligned}
\Delta(A, j, b) &= \Delta_L(A, j, b) + \Delta_R(A, j, b) \\
\widehat{\Delta}(A, j, b) &= \widehat{\Delta}_L(A, j, b) + \widehat{\Delta}_R(A, j, b)
\end{aligned}
\tag{31}
$$

Define the events $\mathcal{E}_1$ and $\mathcal{E}_2$:

$$
\mathcal{E}_1 := \left\{ \sup_{A \in \mathcal{A}_{p, d}} \sqrt{\mathbb{P}(X \in A)} \left| \mathbb{E}(f^*(X) | X \in A) - \bar{y}_{\mathcal{I}_A} \right| \le 20 U \sqrt{\bar{t}(\delta/36)} \right\}
$$

$$
\mathcal{E}_2 := \left\{ \sup_{A \in \mathcal{A}_{p, d}} \left| \sqrt{\mathbb{P}(X \in A)} - \sqrt{|\mathcal{I}_A|/n} \right| \le 5 \sqrt{\bar{t}(\delta/12)} \right\}
$$

Then by Lemmas A.1 and A.2, we have $\mathbb{P}(\mathcal{E}_i) \ge 1 - \delta/3$ for $i = 1, 2$, so we have $\mathbb{P}(\cap_{i=1}^2 \mathcal{E}_i) \ge 1 - \delta$. Below we prove (29) and (30) conditioned on the events $\mathcal{E}_1$ and $\mathcal{E}_2$.

Note that

$$
\begin{aligned}
\sqrt{\Delta_L(A, j, a)} &= \sqrt{\mathbb{P}(X \in A_L)} \left| \mathbb{E}(f^*(X) | X \in A) - \mathbb{E}(f^*(X) | X \in A_L) \right| \\
&\le \sqrt{\mathbb{P}(X \in A_L)} \left| \mathbb{E}(f^*(X) | X \in A) - \bar{y}_{\mathcal{I}_A} \right| + \sqrt{\mathbb{P}(X \in A_L)} \left| \bar{y}_{\mathcal{I}_A} - \bar{y}_{\mathcal{I}_{A_L}} \right| \\
&\quad + \sqrt{\mathbb{P}(X \in A_L)} \left| \bar{y}_{\mathcal{I}_{A_L}} - \mathbb{E}(f^*(X) | X \in A_L) \right| \\
&:= J_1 + J_2 + J_3
\end{aligned}
\tag{32}
$$

To bound $J_1$, we have

$$
J_1 \le \sqrt{\mathbb{P}(X \in A)} \left| \mathbb{E}(f^*(X) | X \in A) - \bar{y}_{\mathcal{I}_A} \right| \le 20 U \sqrt{\bar{t}(\delta/36)}
\tag{33}
$$

where the second inequality is by event $\mathcal{E}_1$. Similarly, to bound $J_3$, we have

$$
J_3 = \sqrt{\mathbb{P}(X \in A_L)} \left| \bar{y}_{\mathcal{I}_{A_L}} - \mathbb{E}(f^*(X) | X \in A_L) \right| \le 20 U \sqrt{\bar{t}(\delta/36)}
\tag{34}
$$

To bound $J_2$, note that

$$J_2 \leq \left| \sqrt{\mathbb{P}(X \in A_L)} - \sqrt{|\mathcal{I}_{A_L}|/n} \right| \cdot |\bar{y}_{\mathcal{I}_A} - \bar{y}_{\mathcal{I}_{A_L}}| + \sqrt{|\mathcal{I}_{A_L}|/n} \cdot |\bar{y}_{\mathcal{I}_A} - \bar{y}_{\mathcal{I}_{A_L}}|$$

$$\leq 5\sqrt{\bar{t}(\delta/12)} \cdot 2U + \sqrt{|\mathcal{I}_{A_L}|/n} \cdot |\bar{y}_{\mathcal{I}_A} - \bar{y}_{\mathcal{I}_{A_L}}|$$
(35)

where the second inequality made use of the event $\mathcal{E}_2$. Combining (32) – (35), we have

$$\sqrt{\Delta_L(A, j, b)} \leq 40U\sqrt{\bar{t}(\delta/36)} + 10U\sqrt{\bar{t}(\delta/12)} + \sqrt{|\mathcal{I}_{A_L}|/n} \cdot |\bar{y}_{\mathcal{I}_A} - \bar{y}_{\mathcal{I}_{A_L}}|$$

$$\leq 50U\sqrt{\bar{t}(\delta/36)} + \sqrt{|\mathcal{I}_{A_L}|/n} \cdot |\bar{y}_{\mathcal{I}_A} - \bar{y}_{\mathcal{I}_{A_L}}|$$

which implies (by Young's inequality)

$$\Delta_L(A, j, a) \leq (1 + 1/\alpha) \cdot 2500U^2\bar{t}(\delta/36) + (1 + \alpha)\frac{|\mathcal{I}_{A_L}|}{n}|\bar{y}_{\mathcal{I}_A} - \bar{y}_{\mathcal{I}_{A_L}}|^2$$
(36)

By a similar argument, we have

$$\Delta_R(A, j, a) \leq (1 + 1/\alpha) \cdot 2500U^2\bar{t}(\delta/36) + (1 + \alpha)\frac{|\mathcal{I}_{A_R}|}{n}|\bar{y}_{\mathcal{I}_A} - \bar{y}_{\mathcal{I}_{A_R}}|^2$$
(37)

Summing up (36) and (37), and by (31), we have

$$\Delta(A, j, a) \leq (1 + 1/\alpha) \cdot 5000U^2\bar{t}(\delta/36) + (1 + \alpha)\widehat{\Delta}(A, j, a)$$

This completes the proof of (29). The proof of (30) is by a similar argument.

$\square$

Lemma A.4 provides upper bounds between $\Delta(A, j, b)$ and $\widehat{\Delta}(A, j, b)$, which serves as a link to translate the population impurity decrease to sample impurity decrease. With all these technical lemmas at hand, we are ready to present the proof Theorem 2.3, as shown in the next subsection.

## A.3 Completing the proof of Theorem 2.3

Define events

$$\mathcal{E}_1 := \left\{ \sup_{A \in \mathcal{A}_{p,d}} \sqrt{\mathbb{P}(X \in A)} \left| \mathbb{E}(f^*(X)|X \in A) - \bar{y}_{\mathcal{I}_A} \right| \leq 20U\sqrt{\bar{t}(\delta/24)} \right\}$$

$$\mathcal{E}_2 := \left\{ \Delta(A, j, a) \leq (1 + \alpha)\widehat{\Delta}(A, j, a) + (1 + 1/\alpha) \cdot 5000U^2\bar{t}(\delta/72) \quad \forall A \in \mathcal{A}_{p,d-1}, \, j \in [p], \, a \in \mathbb{R} \right\}$$

$$\mathcal{E}_3 := \left\{ \widehat{\Delta}(A, j, a) \leq (1 + \alpha)\Delta(A, j, a) + (1 + 1/\alpha) \cdot 5000U^2\bar{t}(\delta/72) \quad \forall A \in \mathcal{A}_{p,d-1}, \, j \in [p], \, a \in \mathbb{R} \right\}$$

Then by Lemmas A.1 and A.4, and note that from the statement of Theorem 2.3, $\bar{t}_2(\delta/72) < 3/4$, so we have $\mathbb{P}(\mathcal{E}_1) \geq 1 - \delta/2$ and $\mathbb{P}(\mathcal{E}_2 \cup \mathcal{E}_3) \geq 1 - \delta/2$, which implies $\mathbb{P}(\cup_{i=1}^3 \mathcal{E}_i) \geq 1 - \delta$. In the following, we prove (10) using a deterministic argument conditioned on $\cup_{i=1}^3 \mathcal{E}_i$.

For any $k \in [d]$ and any leave node $t$ of $\widehat{f}^{(k)}$ (recall that $\widehat{f}^{(k)}$ is the decision tree by CART with depth $k$), let $A_t^{(k)}$ be the corresponding cube, that is, for any $x \in \mathbb{R}^p$, $x \in A_t^{(k)}$ if and only if $x$ is routed to $t$ in $\widehat{f}^{(k)}$. Let $\mathcal{L}^{(k)}$ be the set of all leave nodes of $\widehat{f}^{(k)}$. Then we have

$$\widehat{f}^{(k)}(x) = \sum_{t \in \mathcal{L}^{(k)}} \bar{y}_{\mathcal{I}_{A_t^{(k)}}} 1_{\{x \in A_t^{(k)}\}}$$
(38)

Define a function

$$\widetilde{f}^{(k)}(x) := \sum_{t \in \mathcal{L}^{(k)}} \mathbb{E}\left( f^*(X) \Big| X \in A_t^{(k)}, \mathcal{X}_1^n \right) \cdot 1_{\{x \in A_t^{(k)}\}}$$
(39)

where $\mathcal{X}_1^n$ is the set of iid random variables $\{x_1, ..., x_n\}$, and $X$ is a random variable having the same distribution as $x_1$ but independent of $\mathcal{X}_1^n$. In other words, $\widetilde{f}^{(k)}$ is a tree with the same splitting structure as $\widehat{f}^{(k)}$ and replaces the prediction value of each leave node as the populational conditional mean of $f^*(\cdot)$.

First, using Cauchy-Schwarz inequality, we have

$$\|\widehat{f}^{(k)} - f^*\|_{L^2(X)}^2 \leq 2\|f^* - \widetilde{f}^{(k)}\|_{L^2(X)}^2 + 2\|\widetilde{f}^{(k)} - \widehat{f}^{(k)}\|_{L^2(X)}^2 := 2J_1(k) + 2J_2(k)$$
(40)

To bound $J_1(d)$, we derive recursive inequalities between $J_1(k)$ and $J_1(k+1)$ for all $0 \leq k \leq d - 1$. Note that

$$J_1(k) = \mathbb{E}\left( (f^*(X) - \widetilde{f}^{(k)}(X))^2 \Big| \mathcal{X}_1^n \right)$$

$$= \sum_{t \in \mathcal{L}^{(k)}} \mathbb{P}(X \in A_t | \mathcal{X}_1^n) \cdot \text{Var}(f^*(X)|X \in A_t, \mathcal{X}_1^n)$$
(41)

For each $t \in \mathcal{L}^{(k)}$, let $t_L$ and $t_R$ be the two children of $t$, then we have

$$\mathbb{P}(X \in A_t | \mathcal{X}_1^n) \cdot \text{Var}(f^*(X) | X \in A_t, \mathcal{X}_1^n)$$
$$= \mathbb{P}(X \in A_{t_L} | \mathcal{X}_1^n) \cdot \text{Var}(f^*(X) | X \in A_{t_L}, \mathcal{X}_1^n) \tag{42}$$
$$+ \mathbb{P}(X \in A_{t_R} | \mathcal{X}_1^n) \cdot \text{Var}(f^*(X) | X \in A_{t_R}, \mathcal{X}_1^n) + \Delta(A_t, \hat{j}_t, \hat{b}_t)$$

where

$$(\hat{j}_t, \hat{b}_t) \in \underset{j \in [p], b \in \mathbb{R}}{\text{argmax}} \, \widehat{\Delta}(A_t, j, b)$$

Let us define

$$(j_t, b_t) \in \underset{j \in [p], b \in \mathbb{R}}{\text{argmax}} \, \Delta(A_t, j, b)$$

Then we have

$$\Delta(A_t, \hat{j}_t, \hat{b}_t) \geq \frac{1}{1+\alpha} \widehat{\Delta}(A_t, \hat{j}_t, \hat{b}_t) - (5000/\alpha) U^2 \bar{t}(\delta/72)$$
$$\geq \frac{1}{1+\alpha} \widehat{\Delta}(A_t, j_t, b_t) - (5000/\alpha) U^2 \bar{t}(\delta/72) \tag{43}$$
$$\geq \frac{1}{(1+\alpha)^2} \Delta(A_t, j_t, b_t) - \frac{2+\alpha}{\alpha(1+\alpha)} 5000 U^2 \bar{t}(\delta/72)$$

where the first inequality is by event $\mathcal{E}_3$, the second inequality is by the definition of $(\hat{j}_t, \hat{b}_t)$, and the third inequality is because of event $\mathcal{E}_2$. By Assumption 2.2, we have

$$\Delta(A_t, j_t, b_t) = \sup_{j \in [p], b \in \mathbb{R}} \Delta(A_t, j, b) \geq \lambda \cdot \mathbb{P}(X \in A_t | \mathcal{X}_1^n) \text{Var}(f^*(X) | X \in A_t, \mathcal{X}_1^n) \tag{44}$$

Combining (42), (43) and (44), we have

$$\mathbb{P}(X \in A_{t_L} | \mathcal{X}_1^n) \cdot \text{Var}(f^*(X) | X \in A_{t_L}, \mathcal{X}_1^n) + \mathbb{P}(X \in A_{t_R} | \mathcal{X}_1^n) \cdot \text{Var}(f^*(X) | X \in A_{t_R}, \mathcal{X}_1^n)$$
$$\leq \left(1 - \frac{\lambda}{(1+\alpha)^2}\right) \mathbb{P}(X \in A_t | \mathcal{X}_1^n) \cdot \text{Var}(f^*(X) | X \in A_t, \mathcal{X}_1^n) + \frac{2+\alpha}{\alpha(1+\alpha)} 5000 U^2 \bar{t}(\delta/72)$$

Summing up the inequality above for all $t \in \mathcal{L}^{(k)}$, we have

$$J_1(k+1) \leq \left(1 - \frac{\lambda}{(1+\alpha)^2}\right) J_1(k) + 2^k \cdot \frac{2+\alpha}{\alpha(1+\alpha)} 5000 U^2 \bar{t}(\delta/72)$$

Using the inequality above recursively for $k = 0, 1, ...., d-1$, we have

$$J_1(d) \leq \left(1 - \frac{\lambda}{(1+\alpha)^2}\right)^d J_1(0) + \frac{2+\alpha}{\alpha(1+\alpha)} 5000 U^2 \bar{t}(\delta/72) \sum_{k=1}^d 2^{k-1}$$
$$\leq \left(1 - \frac{\lambda}{(1+\alpha)^2}\right)^d \text{Var}(f^*(X)) + 2^d \cdot \frac{2+\alpha}{\alpha(1+\alpha)} 5000 U^2 \bar{t}(\delta/72) \tag{45}$$

To bound $J_2(d)$, we have

$$J_2(d) = \sum_{t \in \mathcal{L}^{(d)}} \mathbb{P}(X \in A_t) \left(\mathbb{E}(f^*(X) | X \in A_t, \mathcal{X}_1^n) - \bar{y}_{\mathcal{I}_{A_t}}\right)^2 \tag{46}$$
$$\leq 2^d \cdot 400 U^2 \bar{t}(\delta/24)$$

where the inequality made use of event $\mathcal{E}_1$.

Using (45) and (46), and recalling (40), we have

$$\|\widehat{f}^{(k)} - f^*\|_{L^2(X)}^2 \leq 2\left(1 - \frac{\lambda}{(1+\alpha)^2}\right)^d \text{Var}(f^*(X)) + 2^{d+1} \cdot \frac{2+\alpha}{\alpha(1+\alpha)} 5000 U^2 \bar{t}(\delta/72)$$
$$+ 2^{d+1} \cdot 400 U^2 \bar{t}(\delta/24)$$
$$\lesssim \text{Var}(f^*(X)) \cdot (1 - \lambda/(1+\alpha)^2)^d + \frac{2+\alpha}{\alpha(1+\alpha)} \frac{2^d(d \log(np) + \log(1/\delta))}{n} U^2 \tag{47}$$
$$\lesssim \text{Var}(f^*(X)) \cdot (1 - \lambda/(1+\alpha)^2)^d + \frac{2^d(d \log(np) + \log(1/\delta))}{\alpha n} U^2$$

This completes the proof of (10). To prove (11), by taking $\alpha = 1/d$ and $d = \lceil \log_2(n)/(1 - \log_2(1-\lambda)) \rceil$, we have

$$\left(1 - \frac{\lambda}{(1+\alpha)^2}\right)^d = (1-\lambda)^d \left(1 + \frac{\lambda}{1-\lambda}(1 - (1+\alpha)^{-2})\right)^d$$
$$= (1-\lambda)^d \left(1 + \frac{\lambda}{1-\lambda} \frac{2/d + 1/d^2}{(1+1/d)^2}\right)^d \lesssim_\lambda (1-\lambda)^d \tag{48}$$

Note that for $s = \log_2(n)/(1 - \log_2(1 - \lambda))$ we have $(1 - \lambda)^s = 2^s/n$, hence by taking $d = \lceil \log_2(n)/(1 - \log_2(1 - \lambda)) \rceil$, we have

$$(1 - \lambda)^d \leq \frac{2^d}{n} \leq 2n^{-1 + \frac{1}{1 - \log_2(1 - \lambda)}} = 2n^{-\phi(\lambda)}. \tag{49}$$

Combining (47), (48) and (49) and note that $\mathrm{Var}(f^*(X)) \leq M < U$, we have

$$\|\widehat{f}^{(k)} - f^*\|^2_{L^2(X)} \lesssim_{\lambda, U} n^{-\phi(\lambda)}(d^2 \log(np) + d \log(1/\delta))$$
$$\lesssim_{\lambda, U} n^{-\phi(\lambda)}(\log^2(n) \log(np) + \log(n) \log(1/\delta))$$

this completes the proof of (11).

## A.4 Proof of Lemma A.1

The main idea of proving Lemma A.1 is to find a proper finite net of the set $\mathcal{A}_{p,d}$, control the gap on this net, and finally prove the result for all $A \in \mathcal{A}_{p,d}$ based on the approximation gap of the net. We need a few auxiliary results. Let $\mathcal{S} := \{0, 1/n, 2/n, ..., (n-1)/n, 1\}$, and define

$$\widetilde{\mathcal{A}}_{p,d} := \left\{ \prod_{j=1}^{p} [\ell_j, u_j] \in \mathcal{A}_{p,d} \,\middle|\, \ell_j, u_j \in \mathcal{S} \text{ for all } j \in [p] \right\}$$

For any $A = \prod_{j=1}^{p} [\ell_j, u_j] \in \mathcal{A}_{p,d}$, define

$$A' = \prod_{j=1}^{p} [\ell'_j, u'_j]$$

where $\ell'_j := \max\{s \in \mathcal{S} \mid s \leq \ell_j\}$, and $u'_j := \min\{s \in \mathcal{S} \mid s \geq u_j\}$. Roughly speaking, $A'$ is the smallest box with all edges in $\mathcal{S}$ that contains $A$. For any $\widetilde{A} = \prod_{j=1}^{p} [\tilde{\ell}_j, \tilde{u}_j] \in \widetilde{\mathcal{A}}_{p,d}$ with $\tilde{u}_j - \tilde{\ell}_j \geq 2/n$ for all $j \in [p]$, define

$$B(\widetilde{A}) := \widetilde{A} \setminus \prod_{j=1}^{p} \left[ \tilde{\ell}_j + (1/n) \cdot 1_{\{\tilde{\ell}_j \neq 0\}} \,,\, \tilde{u}_j - (1/n) \cdot 1_{\{\tilde{u}_j \neq 1\}} \right].$$

and define $\mathcal{B}_{p,d}$ to be the set of all such sets, that is

$$\mathcal{B}_{p,d} := \left\{ B(\widetilde{A}) \,\middle|\, \widetilde{A} = \prod_{j=1}^{p} [\tilde{\ell}_j, \tilde{u}_j] \in \widetilde{\mathcal{A}}_{p,d} \text{ with } \tilde{u}_j - \tilde{\ell}_j \geq 2/n \right\}$$

The following lemma can be easily verified from the definitions of $\widetilde{\mathcal{A}}_{p,d}$ and $\mathcal{B}_{p,d}$.

**Lemma A.5** *(1) For any $A \in \mathcal{A}_{p,d}$, there exists $B \in \mathcal{B}_{p,d}$ such that $A' \setminus A \subseteq B$.*

*(2) $\mathbb{P}(X \in B) \leq 2\bar{\theta}d/n$ for all $B \in \mathcal{B}_{p,d}$.*

*(3) The cardinality*

$$|\mathcal{B}_{p,d}| \leq |\widetilde{\mathcal{A}}_{p,d}| \leq \binom{p}{d}(n+1)^{2d} \leq p^d(n+1)^{2d}$$

Finally, for any $t \geq 0$, we define

$$\mathcal{A}_{p,d}(t) := \left\{ A \in \mathcal{A}_{p,d} \,\middle|\, \mathbb{P}(X \in A) \leq t \right\}, \quad \text{and} \quad \widetilde{\mathcal{A}}_{p,d}(t) := \left\{ A \in \widetilde{\mathcal{A}}_{p,d} \,\middle|\, \mathbb{P}(X \in A) \leq t \right\}$$

**Lemma A.6** *Suppose Assumption 2.1 holds true. Let $z_1, ..., z_n$ be i.i.d. bounded random variables with $|z_1| \leq V < \infty$ almost surely. Assume that for each $i \in [n]$, $z_i$ is independent of $\{x_j\}_{j \neq i}$, but may be dependent on $x_i$. Given any $\delta \in (0, 1)$, with probability at least $1 - \delta$, it holds*

$$\max_{A \in \widetilde{\mathcal{A}}_{p,d} \setminus \widetilde{\mathcal{A}}_{p,d}(\bar{t}_1(\delta))} \frac{1}{\sqrt{\mathbb{P}(X \in A)}} \left| \frac{1}{n} \sum_{i=1}^{n} z_i 1_{\{x_i \in A\}} - \mathbb{E}(z_1 1_{\{x_1 \in A\}}) \right| \leq 2V\sqrt{\bar{t}_1(\delta)}$$

*where $U = M + m$.*

*Proof.* For each fixed $A \in \widetilde{\mathcal{A}}_{p,d} \setminus \widetilde{\mathcal{A}}_{p,d}(\bar{t}_1(\delta))$, note that

$$\left| \mathbb{E}\left( (z_1 1_{\{x_1 \in A\}} - \mathbb{E}(z_1 1_{\{x_1 \in A\}}))^k \right) \right| \leq (2V)^k \mathbb{P}(X \in A) \quad \forall k \geq 2$$

so by Lemma D.1 with $t = 2V\sqrt{\mathbb{P}(X \in A)}\sqrt{\bar{t}_1(\delta)}$, $\gamma^2 = (2V)^2\mathbb{P}(X \in A)$ and $b = 2V$, we have

$$\mathbb{P}\left(\frac{1}{\sqrt{\mathbb{P}(X \in A)}}\left|\frac{1}{n}\sum_{i=1}^{n} z_i 1_{\{x_i \in A\}} - \mathbb{E}(z_1 1_{\{x_1 \in A\}})\right| > 2V\sqrt{\bar{t}_1(\delta)}\right)$$

$$\leq 2\exp\left(-\frac{n}{4}\left(\frac{4V^2\mathbb{P}(X \in A)\bar{t}_1(\delta)}{4V^2\mathbb{P}(X \in A)} \wedge \frac{2V\sqrt{\mathbb{P}(X \in A)\bar{t}_1(\delta)}}{2V}\right)\right)$$

$$\overset{(i)}{=} 2\exp\left(-\frac{n}{4}\bar{t}_1(\delta)\right) = \delta/(p^d(n+1)^{2d})$$

where $(i)$ is because $\mathbb{P}(X \in A) \geq \bar{t}_1(\delta)$ (since $A \in \widetilde{\mathcal{A}}_{p,d} \setminus \widetilde{\mathcal{A}}_{p,d}(\bar{t}_1(\delta))$). As a result, we have

$$\mathbb{P}\left(\max_{A \in \widetilde{\mathcal{A}}_{p,d} \setminus \widetilde{\mathcal{A}}_{p,d}(\bar{t}_1(\delta))} \frac{1}{\sqrt{\mathbb{P}(X \in A)}}\left|\frac{1}{n}\sum_{i=1}^{n} z_i 1_{\{x_i \in A\}} - \mathbb{E}(z_1 1_{\{x_1 \in A\}})\right| > 2V\sqrt{\bar{t}_1(\delta)}\right)$$

$$\leq \sum_{A \in \widetilde{\mathcal{A}}_{p,d} \setminus \widetilde{\mathcal{A}}_{p,d}(\bar{t}_1(\delta))} \mathbb{P}\left(\frac{1}{\sqrt{\mathbb{P}(X \in A)}}\left|\frac{1}{n}\sum_{i=1}^{n} z_i 1_{\{x_i \in A\}} - \mathbb{E}(z_1 1_{\{x_1 \in A\}})\right| > 2V\sqrt{\bar{t}_1(\delta)}\right)$$

$$\leq |\widetilde{\mathcal{A}}_{p,d} \setminus \widetilde{\mathcal{A}}_{p,d}(\bar{t}_1(\delta))| \cdot \delta/(p^d(n+1)^{2d}) \leq \delta$$

where the last inequality makes use of Lemma A.5 (3).

$\square$

**Lemma A.7** *Let $\mathcal{D}$ be a finite collection of measurable subsets of $[0,1]^p$ satisfying $\mathbb{P}(X \in D) \leq \bar{\alpha}$ for all $D \in \mathcal{D}$ (for some constant $\bar{\alpha} \in (0,1)$). Given any $\delta \in (0,1)$, if*

$$w(\bar{\alpha}, \delta) := (e^2\bar{\alpha}) \vee \frac{\log(|\mathcal{D}|/\delta)}{n} \leq 3/4$$

*then with probability at least $1 - \delta$ it holds*

$$\max_{D \in \mathcal{D}}\left\{\frac{1}{n}\sum_{i=1}^{n} 1_{\{x_i \in D\}}\right\} \leq w(\bar{\alpha}, \delta)$$

*Proof.* For any fixed $D \in \mathcal{D}$, denote $\alpha = \mathbb{P}(X \in D)$, then by Lemma D.2, for any $t \in (0, 3/4]$, we have

$$\mathbb{P}\left(\frac{1}{n}\sum_{i=1}^{n} 1_{\{x_i \in D\}} > t\right) \leq \exp\left(-n\left(t\log(t/\alpha) + (1-t)\log\left(\frac{1-t}{1-\alpha}\right)\right)\right)$$

$$\leq \exp\left(-n\left(t\log(t/\alpha) + (1-t)\log(1-t)\right)\right)$$

$$\leq \exp\left(-n\left(t\log(t/\alpha) + (1-t)(-t-t^2)\right)\right)$$

$$= \exp\left(-n\left(t\left(\log(t/\alpha) - 1\right) + t^3\right)\right)$$

$$\leq \exp\left(-nt\left(\log(t/\alpha) - 1\right)\right)$$

where the third inequality makes use of Lemma D.3 and the assumption $t \leq 3/4$. Take $t = w(\bar{\alpha}, \delta)$, and note that

$$\log(w(\bar{\alpha}, \delta)/\alpha) - 1 \geq \log(w(\bar{\alpha}, \delta)/\bar{\alpha}) - 1 \geq \log(e^2) - 1 \geq 1$$

we have

$$\mathbb{P}\left(\frac{1}{n}\sum_{i=1}^{n} 1_{\{x_i \in B\}} > w(\bar{\alpha}, \delta)\right) \leq \exp\left(-nw(\bar{\alpha}, \delta)\right) \leq \delta/|\mathcal{D}|$$

where the last inequality is because of the definition of $w(\bar{\alpha}, \delta)$. Taking the union bound we have

$$\mathbb{P}\left(\max_{D \in \mathcal{D}}\left\{\frac{1}{n}\sum_{i=1}^{n} 1_{\{x_i \in D\}}\right\} > w(\bar{\alpha}, \delta)\right) \leq |\mathcal{D}| \cdot \delta/|\mathcal{D}| = \delta$$

$\square$

**Corollary A.8** *Under Assumption 2.1 and given $\delta \in (0,1)$, suppose $\bar{t}_2(\delta) < 3/4$, then with probability at least $1 - \delta$, it holds*

$$\max_{B \in \mathcal{B}_{p,d}}\left\{\frac{1}{n}\sum_{i=1}^{n} 1_{\{x_i \in B\}}\right\} \leq \bar{t}_2(\delta)$$

*Proof.* Apply Lemma A.7 with $\mathcal{D} = \mathcal{B}_{p,d}$ and $\bar{\alpha} = 2\bar{\theta}d/n$, and note that $|\mathcal{B}_{p,d}| \leq (n+1)^{2d}p^d$ (by Lemma A.5 (3)) and the definition $\bar{t}_2(\delta) = \frac{2\bar{\theta}e^2 d}{n} \vee \frac{\log(p^d(n+1)^{2d}/\delta)}{n}$. $\square$

**Lemma A.9** *Suppose Assumption 2.1 holds true. Let $z_1, ..., z_n$ be i.i.d. bounded random variables with $|Z| \leq V < \infty$ almost surely. Assume that for each $i \in [n]$, $z_i$ is independent of $\{x_j\}_{j \neq i}$, but may be dependent on $x_i$. Given any $\delta \in (0,1)$, suppose $\bar{t}_2(\delta/2) < 3/4$, then with probability at least $1 - \delta$, it holds*

$$\sup_{A \in \mathcal{A}_{p,d} \setminus \mathcal{A}_{p,d}(\bar{t}(\delta/2))} \frac{1}{\sqrt{\mathbb{P}(X \in A)}} \left| \frac{1}{n} \sum_{i=1}^{n} z_i 1_{\{x_i \in A\}} - \mathbb{E}(z_1 1_{\{x_1 \in A\}}) \right| \leq 5V\sqrt{\bar{t}(\delta/2)}. \tag{50}$$

*Proof.* Define events $\mathcal{E}_1$ and $\mathcal{E}_2$:

$$\mathcal{E}_1 := \left\{ \max_{B \in \mathcal{B}_{p,d}} \left\{ \frac{1}{n} \sum_{i=1}^{n} 1_{\{x_i \in B\}} \right\} \leq \bar{t}_2(\delta/2) \right\}$$

$$\mathcal{E}_2 := \left\{ \max_{A \in \widetilde{\mathcal{A}}_{p,d} \setminus \widetilde{\mathcal{A}}_{p,d}(\bar{t}_1(\delta/2))} \frac{1}{\sqrt{\mathbb{P}(X \in A)}} \left| \frac{1}{n} \sum_{i=1}^{n} z_i 1_{\{x_i \in A\}} - \mathbb{E}(z_1 1_{\{x_1 \in A\}}) \right| \leq V\sqrt{\bar{t}_1(\delta/2)} \right\}$$

Then by Lemma A.6 and Corollary A.8, we have $\mathbb{P}(\mathcal{E}_1) \geq 1 - \delta/2$ and $\mathbb{P}(\mathcal{E}_2) \geq 1 - \delta/2$, hence $\mathbb{P}(\mathcal{E}_1 \cap \mathcal{E}_2) \geq 1 - \delta$. Below we prove that when $\mathcal{E}_1$ and $\mathcal{E}_2$ hold true, inequality (50) holds true.

Note that for any $A \in \mathcal{A}_{p,d} \setminus \mathcal{A}_{p,d}(\bar{t}(\delta/2))$,

$$\frac{1}{\sqrt{\mathbb{P}(X \in A)}} \left| \frac{1}{n} \sum_{i=1}^{n} z_i 1_{\{x_i \in A\}} - \mathbb{E}(z_1 1_{\{x_1 \in A\}}) \right|$$

$$\leq \frac{1}{\sqrt{\mathbb{P}(X \in A)}} \left| \frac{1}{n} \sum_{i=1}^{n} z_i 1_{\{x_i \in A\}} - \frac{1}{n} \sum_{i=1}^{n} z_i 1_{\{x_i \in A'\}} \right|$$

$$+ \frac{1}{\sqrt{\mathbb{P}(X \in A)}} \left| \frac{1}{n} \sum_{i=1}^{n} z_i 1_{\{x_i \in A'\}} - \mathbb{E}(z_1 1_{\{x_1 \in A'\}}) \right| \tag{51}$$

$$+ \frac{1}{\sqrt{\mathbb{P}(X \in A)}} \left| \mathbb{E}(z_1 1_{\{x_1 \in A'\}}) - \mathbb{E}(z_1 1_{\{x_1 \in A\}}) \right|$$

$$:= T_1 + T_2 + T_3$$

To bound $T_1$, we have

$$T_1 = \frac{1}{\sqrt{\mathbb{P}(X \in A)}} \left| \frac{1}{n} \sum_{i=1}^{n} z_i 1_{\{x_i \in A' \setminus A\}} \right| \leq \frac{V}{\sqrt{\mathbb{P}(X \in A)}} \left( \frac{1}{n} \sum_{i=1}^{n} 1_{\{x_i \in A' \setminus A\}} \right)$$

$$\leq \frac{V}{\sqrt{\mathbb{P}(X \in A)}} \max_{B \in \mathcal{B}_{p,d}} \left\{ \frac{1}{n} \sum_{i=1}^{n} 1_{\{x_i \in B\}} \right\} \leq \frac{V}{\sqrt{\mathbb{P}(X \in A)}} \bar{t}_2(\delta/2) \leq V\sqrt{\bar{t}_2(\delta/2)} \tag{52}$$

where the second inequality makes use of Lemma A.5 (1), and the third inequality is by $\mathcal{E}_1$.

To bound $T_2$, note that

$$T_2 = \sqrt{\frac{\mathbb{P}(X \in A')}{\mathbb{P}(X \in A)}} \frac{1}{\sqrt{\mathbb{P}(X \in A')}} \left| \frac{1}{n} \sum_{i=1}^{n} z_i 1_{\{x_i \in A'\}} - \mathbb{E}(z_1 1_{\{x_1 \in A'\}}) \right|$$

$$\leq \sqrt{\frac{\mathbb{P}(X \in A')}{\mathbb{P}(X \in A)}} 2V\sqrt{\bar{t}_1(\delta/2)} \tag{53}$$

where the inequality is by event $\mathcal{E}_2$ and because $A' \in \widetilde{\mathcal{A}}_{p,d}$ and $\mathbb{P}(X \in A') \geq \mathbb{P}(X \in A) \geq \bar{t}(\delta/2) \geq \bar{t}_1(\delta/2)$. Note that

$$\mathbb{P}(X \in A' \setminus A) \leq \frac{2\bar{\theta}d}{n} \leq \bar{t}_2(\delta/2) \leq \mathbb{P}(X \in A) \tag{54}$$

where the first inequality is by Lemma A.5 (2); the second inequality is by the definition of $\hat{t}_2(\delta/2)$ in (22); the third inequality is because $A \in \mathcal{A}_{p,d} \setminus \mathcal{A}_{p,d}(\bar{t}(\delta/2))$. As a result of (53) and (54), we have

$$T_2 \leq \sqrt{\frac{\mathbb{P}(X \in A' \setminus A) + \mathbb{P}(X \in A)}{\mathbb{P}(X \in A)}} 2V\sqrt{\bar{t}_1(\delta/2)} \leq 2\sqrt{2}V\sqrt{\bar{t}_1(\delta/2)} \tag{55}$$

To bound $T_3$, note that

$$T_3 = \frac{1}{\sqrt{\mathbb{P}(X \in A)}} \left| \mathbb{E}(z_1 1_{\{x_1 \in A' \setminus A\}}) \right| \leq \frac{V}{\sqrt{\mathbb{P}(X \in A)}} \mathbb{P}(X \in A' \setminus A)$$

$$\leq V \sqrt{\mathbb{P}(X \in A' \setminus A)} \leq V \sqrt{2\bar{\theta}d/n} \leq V \sqrt{\bar{t}_2(\delta/2)} \tag{56}$$

The proof is complete by combining inequalities (51), (52), (55) and (56), and note that

$$2V \sqrt{\bar{t}_2(\delta/2)} + 2\sqrt{2}V \sqrt{\bar{t}_1(\delta/2)} \leq 5V \sqrt{\bar{t}(\delta/2)}.$$

$\square$

Now we are ready to wrap up the proof of Lemma A.1.

**Completing the proof of Lemma A.1**

Define events $\mathcal{E}_1$ and $\mathcal{E}_2$:

$$\mathcal{E}_1 := \left\{ \sup_{A \in \mathcal{A}_{p,d} \setminus \mathcal{A}_{p,d}(\bar{t}(\delta/8))} \frac{1}{\sqrt{\mathbb{P}(X \in A)}} \left| \frac{1}{n} \sum_{i=1}^{n} 1_{\{x_i \in A\}} - \mathbb{P}(X \in A) \right| \leq 5\sqrt{\bar{t}(\delta/8)} \right\}$$

$$\mathcal{E}_2 := \left\{ \sup_{A \in \mathcal{A}_{p,d} \setminus \mathcal{A}_{p,d}(\bar{t}(\delta/8))} \frac{1}{\sqrt{\mathbb{P}(X \in A)}} \left| \frac{1}{n} \sum_{i=1}^{n} y_i 1_{\{x_i \in A\}} - \mathbb{E}(y_1 1_{\{x_1 \in A\}}) \right| \leq 5U\sqrt{\bar{t}(\delta/8)} \right\}$$

Then by Lemma A.9 with $z_i = y_i$ and $z_i = 1$ respectively, we know that $\mathbb{P}(\mathcal{E}_i) \geq 1 - \delta/4$ for all $i = 1, 2$. So we know $\mathbb{P}(\cap_{i=1}^{2} \mathcal{E}_i) \geq 1 - \delta$. Below we prove that inequality (23) is true when $\cap_{i=1}^{2} \mathcal{E}_i$ hold.

Define $a := 100\bar{t}(\delta/8)$. Then it holds

$$\sup_{A \in \mathcal{A}_{p,d}(a)} \sqrt{\mathbb{P}(X \in A)} \left| \mathbb{E}(f^*(X)|X \in A) - \bar{y}_{\mathcal{I}_A} \right| \leq 2U\sqrt{a} = 20U\sqrt{\bar{t}(\delta/8)} \tag{57}$$

On the other hand, for any $A \in \mathcal{A}_{p,d} \setminus \mathcal{A}_{p,d}(a)$, by event $\mathcal{E}_1$, we have

$$\frac{1}{n} \sum_{i=1}^{n} 1_{\{x_i \in A\}} \geq \mathbb{P}(X \in A) - 5\sqrt{\bar{t}(\delta/8)}\sqrt{\mathbb{P}(X \in A)} \geq \frac{1}{2}\mathbb{P}(X \in A) \tag{58}$$

where the second inequality is because $\mathbb{P}(X \in A) \geq a = 100\bar{t}(\delta/8)$. Therefore we know $\sum_{i=1}^{n} 1_{\{x_i \in A\}} > 0$, and we can write

$$\mathbb{E}(f^*(X)|X \in A) - \bar{y}_{\mathcal{I}_A} = \frac{\mathbb{E}(y_1 1_{\{x_1 \in A\}})}{\mathbb{P}(X \in A)} - \frac{\frac{1}{n} \sum_{i=1}^{n} y_i 1_{\{x_i \in A\}}}{\frac{1}{n} \sum_{i=1}^{n} 1_{\{x_i \in A\}}}$$

$$= \frac{1}{\mathbb{P}(X \in A)} \left( \mathbb{E}(y_1 1_{\{x_1 \in A\}}) - \frac{1}{n} \sum_{i=1}^{n} y_i 1_{\{x_i \in A\}} \right)$$

$$+ \frac{\sum_{i=1}^{n} y_i 1_{\{x_i \in A\}}}{\sum_{i=1}^{n} 1_{\{x_i \in A\}} \mathbb{P}(X \in A)} \left( \frac{1}{n} \sum_{i=1}^{n} 1_{\{x_i \in A\}} - \mathbb{P}(X \in A) \right) \tag{59}$$

$$:= H_1(A) + H_2(A)$$

By event $\mathcal{E}_2$, and note that $a \geq \bar{t}(\delta/8)$, we have

$$\sup_{A \in \mathcal{A}_{p,d} \setminus \mathcal{A}_{p,d}(a)} \sqrt{\mathbb{P}(X \in A)} |H_1(A)| \leq 5U\sqrt{\bar{t}(\delta/8)} \tag{60}$$

By event $\mathcal{E}_1$, and note that $a \geq \bar{t}(\delta/8)$, we have

$$\sup_{A \in \mathcal{A}_{p,d} \setminus \mathcal{A}_{p,d}(a)} \sqrt{\mathbb{P}(X \in A)} |H_2(A)| \leq 5U\sqrt{\bar{t}(\delta/8)} \tag{61}$$

Combining (59), (60) and (61), we have

$$\sup_{A \in \mathcal{A}_{p,d} \setminus \mathcal{A}_{p,d}(a)} \sqrt{\mathbb{P}(X \in A)} \left| \mathbb{E}(f^*(X)|X \in A) - \bar{y}_{\mathcal{I}_A} \right| \leq 10U\sqrt{\bar{t}(\delta/8)}$$

Combining the inequality above with (57) we have

$$\sup_{A \in \mathcal{A}_{p,d}} \sqrt{\mathbb{P}(X \in A)} \left| \mathbb{E}(f^*(X)|X \in A) - \bar{y}_{\mathcal{I}_A} \right| \leq 20U\sqrt{\bar{t}(\delta/8)} \leq 20U\sqrt{\bar{t}(\delta/12)}$$

## A.5 Proof of Lemma A.2

Define $a := \bar{t}(\delta/4)$ and $b := a + \frac{2\bar{\theta}d}{n}$. Define events $\mathcal{E}_1$ and $\mathcal{E}_2$:

$$\mathcal{E}_1 := \left\{ \max_{A \in \tilde{\mathcal{A}}_{p,d}(b)} \frac{1}{n} \sum_{i=1}^n 1_{\{x_i \in A\}} \leq (e^2 b) \vee \frac{\log(2(n+1)^{2d} p^d/\delta)}{n} \right\}$$

$$\mathcal{E}_2 := \left\{ \sup_{A \in \mathcal{A}_{p,d} \setminus \mathcal{A}_{p,d}(a)} \frac{1}{\sqrt{\mathbb{P}(X \in A)}} \left| \mathbb{P}(X \in A) - \frac{1}{n} \sum_{i=1}^n 1_{\{x_i \in A\}} \right| \leq 5\sqrt{\bar{t}(\delta/4)} \right\}$$

Then by Lemmas A.7 and A.9, we know that $\mathbb{P}(\mathcal{E}_1) \geq 1 - \delta/2$ and $\mathbb{P}(\mathcal{E}_2) \geq 1 - \delta/2$, so $\mathbb{P}(\mathcal{E}_1 \cap \mathcal{E}_2) \geq 1 - \delta$. Below we prove (24) when $\mathcal{E}_1 \cap \mathcal{E}_2$ holds.

For $A \in \mathcal{A}_{p,d}$, if $\mathbb{P}(X \in A) \leq a$, then $\mathbb{P}(A') \leq a + \frac{2\bar{\theta}d}{n} = b$. So we have

$$\sup_{A \in \mathcal{A}_{p,d}(a)} \frac{1}{n} \sum_{i=1}^n 1_{\{x_i \in A\}} \leq \sup_{A \in \mathcal{A}_{p,d}(a)} \frac{1}{n} \sum_{i=1}^n 1_{\{x_i \in A'\}} \leq \sup_{\tilde{A} \in \tilde{\mathcal{A}}_{p,d}(b)} \frac{1}{n} \sum_{i=1}^n 1_{\{x_i \in \tilde{A}\}}$$

$$\leq \left( e^2 \bar{t}(\delta/4) + 2e^2 \bar{\theta} d/n \right) \vee \frac{\log(2(n+1)^{2d} p^d/\delta)}{n} \tag{62}$$

$$\leq (e^2 + 1)\bar{t}(\delta/4) \leq 25\bar{t}(\delta/4)$$

where the third inequality is by event $\mathcal{E}_1$ and the definition of $b$; the fourth inequality is because

$$\bar{t}(\delta/4) \geq \bar{t}_1(\delta/4) \geq \frac{1}{n} \log(2p^d(n+1)^{2d}/\delta) \quad \text{and} \quad \bar{t}(\delta/4) \geq \bar{t}_2(\delta/4) \geq 2e^2 \bar{\theta} d/n \ .$$

As a result, we have

$$\sup_{A \in \mathcal{A}_{p,d}(a)} \left| \sqrt{\mathbb{P}(X \in A)} - \sqrt{\frac{1}{n} \sum_{i=1}^n 1_{\{x_i \in A\}}} \right|$$

$$\leq \sup_{A \in \mathcal{A}_{p,d}(a)} \max \left\{ \sqrt{\mathbb{P}(X \in A)}, \sqrt{\frac{1}{n} \sum_{i=1}^n 1_{\{x_i \in A\}}} \right\} \leq 5\sqrt{\bar{t}(\delta/4)} \tag{63}$$

where the second inequality made use of (62).

On the other hand,

$$\sup_{A \in \mathcal{A}_{p,d} \setminus \mathcal{A}_{p,d}(a)} \left| \sqrt{\mathbb{P}(X \in A)} - \sqrt{\frac{1}{n} \sum_{i=1}^n 1_{\{x_i \in A\}}} \right|$$

$$= \sup_{A \in \mathcal{A}_{p,d} \setminus \mathcal{A}_{p,d}(a)} \frac{\left| \mathbb{P}(X \in A) - \frac{1}{n} \sum_{i=1}^n 1_{\{x_i \in A\}} \right|}{\sqrt{\mathbb{P}(X \in A)} + \sqrt{\frac{1}{n} \sum_{i=1}^n 1_{\{x_i \in A\}}}} \tag{64}$$

$$\leq \sup_{A \in \mathcal{A}_{p,d} \setminus \mathcal{A}_{p,d}(a)} \frac{1}{\sqrt{\mathbb{P}(X \in A)}} \left| \mathbb{P}(X \in A) - \frac{1}{n} \sum_{i=1}^n 1_{\{x_i \in A\}} \right| \leq 5\sqrt{\bar{t}(\delta/4)}$$

where the last inequality is by event $\mathcal{E}_2$.

Combining (63) and (64) the proof is complete.

# B  Proofs in Section 3

For any interval $E \in \mathcal{E}$ and any univariate function $g$ on $[0,1]$, let $V_g(E)$ be the total variation of $g$ on $E$. For the additive model (14) and a rectangle $A = \prod_{j=1}^p E_j \in \mathcal{A}$, define $V_{f^*}(A) = \sum_{j=1}^p V_{f_j^*}(E_j)$. Recall that $X$ is a random variable with the same distribution as $x_i$, and $X^{(j)}$ is the $j$-th coordinate of $X$.

## B.1  Technical lemmas

**Lemma B.1** *For any rectangle $A \subseteq [0,1]^p$, any $j \in [p]$ and any $b \in \mathbb{R}$, it holds*

$$\Delta(A, j, b) = \left( \mathbb{E}(f^*(X) 1_{\{X \in A_R\}}) - \mathbb{E}(f^*(X)|X \in A)\mathbb{P}(X \in A_R) \right)^2 \frac{\mathbb{P}(X \in A)}{\mathbb{P}(X \in A_L)\mathbb{P}(X \in A_R)}$$

*where $A_L = A_L(j, b)$ and $A_R = A_R(j, b)$.*

*Proof.* We use the notations $\nu := \mathbb{E}(f^*(X)|X \in A)$, $\nu_L := \mathbb{E}(f^*(X)|X \in A_L)$ and $\nu_R := \mathbb{E}(f^*(X)|X \in A_R)$. First, note that

$$\mathbb{E}\left((f^*(X) - \nu)^2 1_{\{X \in A_L\}}\right) = \mathbb{E}\left((f^*(X) - \nu_L + \nu_L - \nu)^2 1_{\{X \in A_L\}}\right)$$
$$= \mathbb{E}\left((f^*(X) - \nu_L)^2 1_{\{X \in A_L\}}\right) + (\nu_L - \nu)^2 \mathbb{P}(X \in A_L) \tag{65}$$

Similarly, we have

$$\mathbb{E}\left((f^*(X) - \nu)^2 1_{\{X \in A_R\}}\right) = \mathbb{E}\left((f^*(X) - \nu_R)^2 1_{\{X \in A_R\}}\right) + (\nu_R - \nu)^2 \mathbb{P}(X \in A_R) \tag{66}$$

Summing up (65) and (66) we have

$$(\nu_L - \nu)^2 \mathbb{P}(X \in A_L) + (\nu_R - \nu)^2 \mathbb{P}(X \in A_R)$$
$$= \mathbb{E}\left((f^*(X) - \nu)^2 1_{\{X \in A\}}\right) - \mathbb{E}\left((f^*(X) - \nu_L)^2 1_{\{X \in A_L\}}\right) - \mathbb{E}\left((f^*(X) - \nu_R)^2 1_{\{X \in A_R\}}\right) \tag{67}$$
$$= \Delta(A, j, b)$$

Note that

$$(\nu_L - \nu)^2 \mathbb{P}(X \in A_L) = \left(\mathbb{E}(f^*(X)1_{\{X \in A_L\}}) - \nu\mathbb{P}(X \in A_L)\right)^2 (\mathbb{P}(X \in A_L))^{-1}$$
$$= \left(\nu\mathbb{P}(X \in A) - \mathbb{E}(f^*(X)1_{\{X \in A_R\}}) - \nu\mathbb{P}(X \in A_L)\right)^2 (\mathbb{P}(X \in A_L))^{-1}$$
$$= \left(\nu\mathbb{P}(X \in A_R) - \mathbb{E}(f^*(X)1_{\{X \in A_R\}})\right)^2 (\mathbb{P}(X \in A_L))^{-1} \tag{68}$$
$$= (\nu_R - \nu)^2 \frac{(\mathbb{P}(X \in A_R))^2}{\mathbb{P}(X \in A_L)}$$

Combining (67) and (68) we have

$$\Delta(A, j, b) = (\nu_R - \nu)^2 \frac{(\mathbb{P}(X \in A_R))^2}{\mathbb{P}(X \in A_L)} + (\nu_R - \nu)^2 \mathbb{P}(X \in A_R) = (\nu_R - \nu)^2 \frac{\mathbb{P}(X \in A_R)\mathbb{P}(X \in A)}{\mathbb{P}(X \in A_L)}$$
$$= \left(\mathbb{E}(f^*(X)1_{\{X \in A_R\}}) - \nu\mathbb{P}(X \in A_R)\right)^2 \frac{\mathbb{P}(X \in A)}{\mathbb{P}(X \in A_L)\mathbb{P}(X \in A_R)}$$

$\square$

**Lemma B.2** *Suppose Assumption 2.1 holds true, and $f^*$ has the additive structure in* (14). *Then for any* $A = \prod_{j=1}^p [\ell_j, u_j] \subseteq [0, 1]^p$, *it holds*

$$\max_{j \in [p], b \in \mathbb{R}} \sqrt{\Delta(A, j, b)} \geq \frac{\sqrt{\mathbb{P}(X \in A)}\text{Var}(f^*(X)|X \in A)}{\sum_{k=1}^p \int_{\ell_k}^{u_k} \sqrt{q_A^{(k)}(t)(1 - q_A^{(k)}(t))}dV_{f_k^*}([\ell_j, t])}$$

*where* $q_A^{(k)}(t) := \mathbb{P}(X^{(k)} \leq t|x_1 \in A)$.

*Proof.* For a fixed $A = \prod_{j=1}^p [\ell_j, u_j] \subseteq [0, 1]^p$, without loss of generality, assume $\mathbb{E}(f^*(X)|X \in A) = 0$. Note that for any $j \in [p]$,

$$\max_{b \in \mathbb{R}} \sqrt{\Delta(A, j, b)} \geq \frac{\int_{\ell_j}^{u_j} \sqrt{q_A^{(j)}(s)(1 - q_A^{(j)}(s))}\sqrt{\Delta(A, j, s)} \, dV_{f_j^*}([\ell_j, s])}{\int_{\ell_j}^{u_j} \sqrt{q_A^{(j)}(s)(1 - q_A^{(j)}(s))} \, dV_{f_j^*}([\ell_j, s])} \tag{69}$$

where $s$ the integration variable. Because $q_A^{(j)}(s) = \mathbb{P}(X \in A_L(j, s))/\mathbb{P}(X \in A)$, using Lemma B.1 and recall that we have assumed $\mathbb{E}(f^*(X)|X \in A) = 0$, we have

$$\int_{\ell_j}^{u_j} \sqrt{q_A^{(j)}(s)(1 - q_A^{(j)}(s))}\sqrt{\Delta(A, j, s)} \, dV_{f_j^*}([\ell_j, s])$$

$$= \frac{1}{\sqrt{\mathbb{P}(X \in A)}} \int_{\ell_j}^{u_j} \left|\mathbb{E}(f^*(X)1_{\{X \in A_R(j, s)\}})\right| dV_{f_j^*}([\ell_j, s])$$

$$= \frac{1}{\sqrt{\mathbb{P}(X \in A)}} \int_{\ell_j}^{u_j} \left|\mathbb{E}(f^*(X)1_{\{X \in A\}}1_{\{X^{(j)} > s\}})\right| dV_{f_j^*}([\ell_j, s])$$

$$\geq \frac{1}{\sqrt{\mathbb{P}(X \in A)}}\left|\int_{\ell_j}^{u_j} \mathbb{E}(f^*(X)1_{\{X \in A\}}1_{\{X^{(j)} > s\}}) \, df_j^*(s)\right|$$

$$= \frac{1}{\sqrt{\mathbb{P}(X \in A)}}\left|\mathbb{E}(f^*(X)(f_j^*(X^{(j)}) - f_j^*(\ell_j))1_{\{X \in A\}})\right|$$

$$= \frac{1}{\sqrt{\mathbb{P}(X \in A)}}\left|\mathbb{E}(f^*(X)f_j^*(X^{(j)})1_{\{X \in A\}})\right|$$

where the last equality makes use of the assumption that $\mathbb{E}(f^*(X)|X \in A) = 0$. Combining the inequality above with (69), we have

$$\max_{b \in \mathbb{R}} \sqrt{\Delta(A, j, b)} \geq \frac{1}{\sqrt{\mathbb{P}(X \in A)}} \frac{\left| \mathbb{E}(f^*(X)f_j^*(X^{(j)})1_{\{X \in A\}}) \right|}{\int_{\ell_j}^{u_j} \sqrt{q_A^{(j)}(s)(1 - q_A^{(j)}(s))} \, dV_{f_j^*}([\ell_j, s])}$$

As a result, we have

$$\max_{j \in [p], b \in \mathbb{R}} \sqrt{\Delta(A, j, b)} \geq \frac{1}{\sqrt{\mathbb{P}(X \in A)}} \frac{\sum_{j=1}^{p} \left| \mathbb{E}(f^*(X)f_j^*(X^{(j)})1_{\{X \in A\}}) \right|}{\sum_{j=1}^{p} \int_{\ell_j}^{u_j} \sqrt{q_A^{(j)}(s)(1 - q_A^{(j)}(s))} \, dV_{f_j^*}([\ell_j, s])} \tag{70}$$

By the additive structure (14) we have

$$\sum_{j=1}^{p} \left| \mathbb{E}(f^*(X)f_j^*(X^{(j)})1_{\{X \in A\}}) \right| \geq \mathbb{E}((f^*(X))^2 1_{\{X \in A\}}) = \mathbb{P}(X \in A)\mathrm{Var}(f^*(X)|X \in A) \tag{71}$$

Combining (70) and (71), the proof is complete. $\qquad\square$

**Lemma B.3** *Suppose Assumption 2.1 holds true, and $f^*$ has the additive structure in (14). If for any $A = \prod_{j=1}^{p}[\ell_j, u_j] \subseteq [0,1]^p$ and any $k \in [p]$ it holds*

$$\left( \int_{\ell_k}^{u_k} \sqrt{q_A^{(k)}(t)(1 - q_A^{(k)}(t))} \, dV_{f_k^*}([\ell_k, t]) \right)^2 \leq \frac{\tau^2}{u_k - \ell_k} \inf_{w \in \mathbb{R}} \int_{\ell_k}^{u_k} |f_k^*(t) - w|^2 \, dt \tag{72}$$

*Then Assumption 2.2 is satisfied with $\lambda = \underline{\theta}/(p\tau^2\bar{\theta})$.*

*Proof.* Given $A = \prod_{j=1}^{p}[\ell_j, u_j] \subseteq [0,1]^p$, without loss of generality, assume $\mathbb{E}(f^*(X)|X \in A) = 0$. Let $p_X(\cdot)$ be the density of $X$ on $[0,1]^p$. Then we have

$$\mathrm{Var}(f^*(X)|X \in A) = \frac{1}{\mathbb{P}(X \in A)} \int_A (f^*(z))^2 p_X(z) \, dz \geq \frac{\underline{\theta}}{\mathbb{P}(X \in A)} \int_A (f^*(z))^2 \, dz \tag{73}$$

where the second inequality made use of Assumption 2.1 $(i)$. Denote $c_j := \frac{1}{u_j - \ell_j} \int_{\ell_j}^{u_j} f_j^*(t)dt$ and $c := \sum_{j=1}^{p} c_j$, then we have

$$\int_A (f^*(z))^2 \, dz = \int_A \left( c + \sum_{j=1}^{p} f_j^*(z_j) - c_j \right)^2 dz_1 dz_2 \cdots dz_p$$

$$= \int_A c^2 + \sum_{j=1}^{p} (f_j^*(z_j) - c_j)^2 \, dz_1 dz_2 \cdots dz_p$$

$$\geq \sum_{j=1}^{p} \frac{\prod_{k=1}^{p}(u_k - \ell_k)}{u_j - \ell_j} \int_{\ell_j}^{u_j} (f_j^*(t) - c_j)^2 \, dt$$

$$\geq \frac{\mathbb{P}(X \in A)}{\bar{\theta}} \sum_{j=1}^{p} \frac{1}{u_j - \ell_j} \int_{\ell_j}^{u_j} (f_j^*(t) - c_j)^2 \, dt$$

Combining the inequality above with (73) we have

$$\mathrm{Var}(f^*(X)|X \in A) \geq \frac{\underline{\theta}}{\bar{\theta}} \sum_{j=1}^{p} \frac{1}{u_j - \ell_j} \int_{\ell_j}^{u_j} (f_j^*(t) - c_j)^2 \, dt \tag{74}$$

We use $H_k^2$ to denote the LHS of (72), then (72) implies

$$\frac{1}{u_j - \ell_j} \int_{\ell_j}^{u_j} |f_j^*(t) - c_j|^2 \, dt \geq \frac{1}{\tau^2} H_j^2 \tag{75}$$

As a result of (74) and (75), we have

$$\mathrm{Var}(f^*(X)|X \in A) \geq \frac{\underline{\theta}}{\bar{\theta}\tau^2} \sum_{j=1}^{p} H_k^2 \tag{76}$$

By Lemma B.2 we have

$$
\begin{aligned}
\max_{j \in [p], b \in \mathbb{R}} \Delta(A, j, b) &\geq \frac{\mathbb{P}(X \in A)\mathrm{Var}(f^*(X)|X \in A)^2}{(\sum_{k=1}^p H_k)^2} \\
&\geq \frac{\theta}{\bar{\theta}\tau^2} \frac{\sum_{j=1}^p H_k^2}{(\sum_{k=1}^p H_k)^2} \mathbb{P}(X \in A)\mathrm{Var}(f^*(X)|X \in A) \\
&\geq \frac{\theta}{p\bar{\theta}\tau^2} \mathbb{P}(X \in A)\mathrm{Var}(f^*(X)|X \in A)
\end{aligned}
$$

where the second inequality is by (76), and the last inequality made use of the Cauchy-Schwarz inequality.

$\square$

## B.2 Proof of Proposition 3.1

For any $A = \prod_{j=1}^p [\ell_j, u_j] \subseteq [0,1]^p$ and any $k \in [p]$ it holds

$$
\begin{aligned}
\left( \int_{\ell_k}^{u_k} \sqrt{q_A^{(k)}(t)(1 - q_A^{(k)}(t))} \, dV_{f_k^*}([\ell_k, t]) \right)^2 &\leq \frac{1}{4} \left( \int_{\ell_k}^{u_k} |(f_k^*)'(t)| \, dt \right)^2 \\
&\leq \frac{\tau^2/4}{u_k - \ell_k} \inf_{w \in \mathbb{R}} \int_{\ell_k}^{u_k} |f_k^*(t) - w|^2 \, dt
\end{aligned}
$$

where the first inequality is by Cauchy-Schwarz inequality, and the second is because $f_k^* \in LRP([0,1], \tau)$. Using Lemma B.3, the proof of complete.

## B.3 Proof of Proposition 3.2

For any $A = \prod_{j=1}^p [\ell_j, u_j] \subseteq [0,1]^p$ and any $k \in [p]$, we prove that

$$
\left( \int_{\ell_k}^{u_k} \sqrt{q_A^{(k)}(t)(1 - q_A^{(k)}(t))} \, dV_{f_k^*}([\ell_k, t]) \right)^2 \leq \max\left\{ \frac{2r\bar{\theta}}{\theta}, \frac{r^2}{2\alpha} \right\} \frac{\max\{9\beta^2, 32 + \beta^2\}}{u_k - \ell_k} \inf_{w \in \mathbb{R}} \int_{\ell_k}^{u_k} |f_k^*(t) - w|^2 \, dt \tag{77}
$$

Then the conclusion follows Lemma B.3.

For fixed $A$ and $k \in [p]$, to simplify the notation, we denote $g := f_k^*$, $a := \ell_k$, $b := u_k$, $q(t) := q_A^{(k)}(t)$ for all $t \in [\ell_k, u_k]$, and $t_j := t_j^{(k)}$ for $j = 0, 1, ..., r$. Then (77) can be written as

$$
\left( \int_a^b \sqrt{q(t)(1 - q(t))} \, dV_g([a, t]) \right)^2 \leq 2r \max\left\{ \frac{\bar{\theta}}{\theta}, \frac{r}{4\alpha} \right\} \frac{\max\{9\beta^2, 32 + \beta^2\}}{b - a} \inf_{w \in \mathbb{R}} \int_a^b (g(t) - w)^2 \, dt \tag{78}
$$

For any $s \in (0, 1)$, define $\Delta g(s) := \lim_{t \to s+} g(t) - \lim_{t \to s-} g(t)$. Let $j', j'' \in [r]$ such that $t_{j'-1} \leq a < t_{j'}$ and $t_{j''-1} < b \leq t_{j''}$, and define $r' = j'' - j' + 1$, and

$$
z_0 = a, \; z_1 = t_{j'}, z_2 = t_{j'+1}, \; ..., z_{r'-1} = t_{j''-1}, \; z_{r'} = b.
$$

Then we have

$$
\begin{aligned}
&\left( \int_a^b \sqrt{q(t)(1 - q(t))} \, \mathrm{d}V_g([a, t]) \right)^2 \\
=&\left( \sum_{j=1}^{r'} \int_{z_{j-1}}^{z_j} \sqrt{q(t)(1 - q(t))} |g'(t)| \, \mathrm{d}t + \sum_{j=1}^{r'-1} \sqrt{q(z_j)(1 - q(z_j))} \Delta g(z_j) \right)^2 \\
\leq& 2r' \sum_{j=1}^{r'} \left( \int_{z_{j-1}}^{z_j} \sqrt{q(t)(1 - q(t))} |g'(t)| \, \mathrm{d}t \right)^2 + 2(r' - 1) \sum_{j=1}^{r'-1} q(z_j)(1 - q(z_j)) |\Delta g(z_j)|^2
\end{aligned} \tag{79}
$$

We have the following 4 claims bounding the terms in the last line of the display above.

**Claim B.4** *For $j \in \{1, r'\}$, it holds*

$$
\left( \int_{z_{j-1}}^{z_j} \sqrt{q(t)(1 - q(t))} |g'(t)| \, \mathrm{d}t \right)^2 \leq \frac{\bar{\theta}\beta^2}{\theta(b - a)} \inf_{w \in \mathbb{R}} \int_{z_{j-1}}^{z_j} (g(t) - w)^2 \mathrm{d}t
$$

*Proof of Claim B.4*: We just prove the claim for $j = 1$. The proof for $j = r'$ follows a similar argument. To prove the claim for $j = 1$, we discuss two cases.

(Case 1) $q(z_1) \leq 1/2$. Then we have $\sqrt{q(t)(1 - q(t))} \leq \sqrt{q(z_1)(1 - q(z_1))}$, hence

$$\left( \int_a^{z_1} \sqrt{q(t)(1 - q(t))} |g'(t)| \, dt \right)^2 \leq q(z_1)(1 - q(z_1)) \left( \int_a^{z_1} |g'(t)| \, dt \right)^2$$

$$\leq q(z_1)(1 - q(z_1)) \frac{\beta^2}{z_1 - a} \inf_{w \in \mathbb{R}} \int_a^{z_1} (g(t) - w)^2 \, dt \qquad (80)$$

$$\leq \frac{\bar{\theta}\beta^2}{\underline{\theta}(b - a)} \inf_{w \in \mathbb{R}} \int_a^{z_1} (g(t) - w)^2 dt$$

where the second inequality is because $g \in LRP((a, z_1), \beta)$; and the last inequality makes use of the fact $q(z_1) \leq \bar{\theta}(z_1 - a)/(\underline{\theta}(b - a))$.

(Case 2) $q(z_1) > 1/2$. Then we have

$$z_1 - a \geq \frac{\underline{\theta}(b - a)}{\bar{\theta}} q(z_1) \geq \frac{\underline{\theta}(b - a)}{2\bar{\theta}} \qquad (81)$$

As a result,

$$\left( \int_a^{z_1} \sqrt{q(t)(1 - q(t))} |g'(t)| \, dt \right)^2 \leq \frac{1}{4} \left( \int_a^{z_1} |g'(t)| \, dt \right)^2 \leq \frac{\beta^2}{4(z_1 - a)} \inf_{w \in \mathbb{R}} \int_a^{z_1} (g(t) - w)^2 \, dt$$

$$\leq \frac{\bar{\theta}\beta^2}{2\underline{\theta}(b - a)} \inf_{w \in \mathbb{R}} \int_a^{z_1} (g(t) - w)^2 \, dt$$

where the first inequality is by Cauchy-Schwarz inequality; the second inequality is because $g \in LRP((a, z_1), \beta)$; the third inequality is by (81).

Combining (Caes 1) and (Case 2), the proof of Claim B.4 is complete.

$\square$

**Claim B.5** *For $j \in \{1, r' - 1\}$, it holds*

$$q(z_j)(1 - q(z_j))|\Delta g(z_j)|^2 \leq \max \left\{ \frac{4\bar{\theta}}{\underline{\theta}}, \frac{r}{\alpha} \right\} \frac{\max\{\beta^2, 4\}}{b - a} \inf_{w \in \mathbb{R}} \int_{z_0}^{z_2} (g(t) - w)^2 \, dt$$

*Proof of Claim B.5*: We just prove the claim for $j = 1$. The proof for $j = r' - 1$ follows a similar argument. To prove the claim for $j = 1$, we discuss two cases.

(Case 1) $|\Delta g(z_1)| > 4 \max\{\int_{z_0}^{z_1} |g'(t)| \, dt, \int_{z_1}^{z_2} |g'(t)| \, dt\}$. Then by Lemma D.6 we have

$$\inf_{w \in \mathbb{R}} \int_{z_0}^{z_2} (g(t) - w)^2 \, dt \geq \min \left\{ z_1 - z_0, z_2 - z_1 \right\} \cdot \frac{(\Delta g(z_1))^2}{16} \qquad (82)$$

Note that

$$\min \left\{ z_1 - z_0, z_2 - z_1 \right\} \geq \min \left\{ \frac{\underline{\theta} q(z_1)}{\bar{\theta}}, \frac{\alpha}{r}(b - a) \right\} \geq \min \left\{ \frac{\underline{\theta} q(z_1)}{\bar{\theta}}, \frac{\alpha}{r} \right\}(b - a) \qquad (83)$$

So by (82) and (83) we have

$$|\Delta g(z_1)|^2 \leq \max \left\{ \frac{\bar{\theta}}{\underline{\theta} q(z_1)}, \frac{r}{\alpha} \right\} \frac{16}{b - a} \inf_{w \in \mathbb{R}} \int_{z_0}^{z_2} (g(t) - w)^2 \, dt$$

As a result,

$$q(z_1)(1 - q(z_1))|\Delta g(z_1)|^2$$

$$\leq \max \left\{ \frac{\bar{\theta}}{\underline{\theta}}(1 - q(z_1)), \frac{r}{\alpha} q(z_1)(1 - q(z_1)) \right\} \frac{16}{b - a} \inf_{w \in \mathbb{R}} \int_{z_0}^{z_2} (g(t) - w)^2 \, dt$$

$$\leq \max \left\{ \frac{\bar{\theta}}{\underline{\theta}}, \frac{r}{4\alpha} \right\} \frac{16}{b - a} \inf_{w \in \mathbb{R}} \int_{z_0}^{z_2} (g(t) - w)^2 \, dt$$

where the second inequality is by Cauchy-Schwarz inequality.

(Case 2) $|\Delta g(z_1)| \leq 4 \max\{\int_{z_0}^{z_1} |g'(t)| \, dt, \int_{z_1}^{z_2} |g'(t)| \, dt\}$. Then we have

$$q(z_1)(1 - q(z_1))|\Delta g(z_1)|^2 \leq 4q(z_1)(1 - q(z_1)) \max \left\{ \int_{z_0}^{z_1} |g'(t)| \, dt, \int_{z_1}^{z_2} |g'(t)| \, dt \right\}^2 \qquad (84)$$

By the same argument in (80), we have

$$4q(z_1)(1 - q(z_1))\Big(\int_{z_0}^{z_1} |g'(t)| \, \mathrm{d}t\Big)^2 \leq \frac{4\bar{\theta}\beta^2}{\underline{\theta}(b-a)} \inf_{w \in \mathbb{R}} \int_{z_0}^{z_1} (g(t) - w)^2 \mathrm{d}t \tag{85}$$

On the other hand,

$$4q(z_1)(1 - q(z_1))\Big(\int_{z_1}^{z_2} |g'(t)| \, \mathrm{d}t\Big)^2 \leq \Big(\int_{z_1}^{z_2} |g'(t)| \, \mathrm{d}t\Big)^2$$

$$\leq \frac{\beta^2}{z_2 - z_1} \inf_{w \in \mathbb{R}} \int_{z_1}^{z_2} (g(t) - w)^2 \, \mathrm{d}t \leq \frac{r\beta^2}{\alpha(b-a)} \inf_{w \in \mathbb{R}} \int_{z_1}^{z_2} (g(t) - w)^2 \mathrm{d}t \tag{86}$$

where the second inequality is because $g \in LRP((z_1, z_2), \beta)$; the last inequality is because $z_2 - z_1 \geq \alpha/r \geq (\alpha/r)(b - a)$. By (84), (85) and (86), we have

$$q(z_1)(1 - q(z_1))|\Delta g(z_1)|^2 \leq \max\Big\{\frac{4\bar{\theta}}{\underline{\theta}}, \frac{r}{\alpha}\Big\} \frac{\beta^2}{b-a} \inf_{w \in \mathbb{R}} \int_{z_0}^{z_2} (g(t) - w)^2 \, \mathrm{d}t$$

Combining (Case 1) and (Case 2), the proof of Claim B.5 is complete.

$$\square$$

**Claim B.6** *For $2 \leq j \leq r' - 1$, it holds*

$$\Big(\int_{z_{j-1}}^{z_j} \sqrt{q(t)(1 - q(t))}|g'(t)| \, \mathrm{d}t\Big)^2 \leq \frac{r\beta^2}{4\alpha} \inf_{w \in \mathbb{R}} \int_{z_{j-1}}^{z_j} (g(t) - w)^2 \mathrm{d}t$$

*Proof of Claim B.6*: Note that

$$\Big(\int_{z_{j-1}}^{z_j} \sqrt{q(t)(1 - q(t))}|g'(t)| \, \mathrm{d}t\Big)^2$$

$$\leq \frac{1}{4}\Big(\int_{z_{j-1}}^{z_j} |g'(t)| \, \mathrm{d}t\Big)^2 \leq \frac{\beta^2}{4(z_j - z_{j-1})} \inf_{w \in \mathbb{R}} \int_{z_{j-1}}^{z_j} (g(t) - w)^2 \mathrm{d}t$$

$$\leq \frac{r\beta^2}{4\alpha(b-a)} \inf_{w \in \mathbb{R}} \int_{z_{j-1}}^{z_j} (g(t) - w)^2 \mathrm{d}t$$

where the first inequality is by Cauchy-Schwarz inequality; the second inequality is because $g \in LRP((z_{j-1}, z_j), \beta)$; the last inequality is by the assumption that $t_j - t_{j-1} \geq \alpha/r$.

$$\square$$

**Claim B.7** *For $2 \leq j \leq r' - 2$, it holds*

$$q(z_j)(1 - q(z_j))|\Delta g(z_j)|^2 \leq \frac{r\max\{\beta^2, 4\}}{\alpha} \inf_{w \in \mathbb{R}} \int_{z_{j-1}}^{z_{j+1}} (g(t) - w)^2 \, \mathrm{d}t$$

*Proof of Claim B.7*: We discuss two cases.

(Case 1) $|\Delta g(z_j)| > 4 \max\{\int_{z_{j-1}}^{z_j} |g'(t)| \, \mathrm{d}t, \int_{z_j}^{z_{j+1}} |g'(t)| \, \mathrm{d}t\}$. Then by Lemma D.6 we have

$$\inf_{w \in \mathbb{R}} \int_{z_{j-1}}^{z_{j+1}} (g(t) - w)^2 \, \mathrm{d}t \geq \min\{z_j - z_{j-1}, z_{j+1} - z_j\} \cdot \frac{(\Delta g(z_j))^2}{16} \geq \frac{\alpha}{r} \frac{(\Delta g(z_j))^2}{16}$$

As a result,

$$q(z_j)(1 - q(z_j))|\Delta g(z_j)|^2 \leq \frac{16r}{\alpha} q(z_j)(1 - q(z_j)) \inf_{w \in \mathbb{R}} \int_{z_{j-1}}^{z_{j+1}} (g(t) - w)^2 \, \mathrm{d}t$$

$$\leq \frac{4r}{\alpha} \inf_{w \in \mathbb{R}} \int_{z_{j-1}}^{z_{j+1}} (g(t) - w)^2 \, \mathrm{d}t$$

(Case 2) $|\Delta g(z_j)| \le 4 \max\{\int_{z_{j-1}}^{z_j} |g'(t)|\,\mathrm{d}t\,,\int_{z_j}^{z_{j+1}} |g'(t)|\,\mathrm{d}t\}$. Then we have

$$
q(z_j)(1-q(z_j))|\Delta g(z_j)|^2
$$

$$
\le 4q(z_j)(1-q(z_j)) \max\left\{ \int_{z_{j-1}}^{z_j} |g'(t)|\,\mathrm{d}t\,, \int_{z_j}^{z_{j+1}} |g'(t)|\,\mathrm{d}t \right\}^2
$$

$$
\le \max\left\{ \int_{z_{j-1}}^{z_j} |g'(t)|\,\mathrm{d}t\,, \int_{z_j}^{z_{j+1}} |g'(t)|\,\mathrm{d}t \right\}^2
$$

$$
\le \max\left\{ \frac{\beta^2}{z_j - z_{j-1}} \inf_{w\in\mathbb{R}} \int_{z_{j-1}}^{z_j} (g(t)-w)^2\,\mathrm{d}t,\ \frac{\beta^2}{z_{j+1}-z_j} \inf_{w\in\mathbb{R}} \int_{z_j}^{z_{j+1}} (g(t)-w)^2\,\mathrm{d}t \right\}
$$

$$
\le \frac{\beta^2 r}{\alpha} \max\left\{ \inf_{w\in\mathbb{R}} \int_{z_{j-1}}^{z_j} (g(t)-w)^2\,\mathrm{d}t,\ \inf_{w\in\mathbb{R}} \int_{z_j}^{z_{j+1}} (g(t)-w)^2\,\mathrm{d}t \right\}
$$

$$
\le \frac{\beta^2 r}{\alpha} \inf_{w\in\mathbb{R}} \int_{z_{j-1}}^{z_{j+1}} (g(t)-w)^2\,\mathrm{d}t
$$

where the second inequality is by Cauchy-Schwarz inequality; the third inequality is because $g \in LRP((z_{j-1}, z_j), \beta)$ and $g \in LRP((z_j, z_{j+1}), \beta)$.

Combining (Case 1) and (Case 2), and note that $b - a \le 1$, the proof of Claim B.7 is complete.

$\square$

## Completing the proof of Proposition 3.2

By (79) and note that $r' \le r$, we have

$$
\left( \int_a^b \sqrt{q(t)(1-q(t))}\,\mathrm{d}V_g([a,t]) \right)^2
$$

$$
\le 2r \sum_{j=1}^{r'} \left( \int_{z_{j-1}}^{z_j} \sqrt{q(t)(1-q(t))}|g'(t)|\,\mathrm{d}t \right)^2 + 2r \sum_{j=1}^{r'-1} q(z_j)(1-q(z_j))|\Delta g(z_j)|^2 \tag{87}
$$

By Claims B.4 and B.6, we have

$$
\sum_{j=1}^{r'} \left( \int_{z_{j-1}}^{z_j} \sqrt{q(t)(1-q(t))}|g'(t)|\,\mathrm{d}t \right)^2
$$

$$
\le \max\left\{ \frac{\bar\theta \beta^2}{\underline\theta(b-a)}, \frac{r\beta^2}{4\alpha} \right\} \sum_{j=1}^{r'} \inf_{w\in\mathbb{R}} \int_{z_{j-1}}^{z_j} (g(t)-w)^2\,\mathrm{d}t \tag{88}
$$

$$
\le \max\left\{ \frac{\bar\theta}{\underline\theta}, \frac{r}{4\alpha} \right\} \frac{\beta^2}{b-a} \inf_{w\in\mathbb{R}} \int_a^b (g(t)-w)^2\,\mathrm{d}t
$$

By Claims B.5 and B.7, we have

$$
\sum_{j=1}^{r'-1} q(z_j)(1-q(z_j))|\Delta g(z_j)|^2
$$

$$
\le \max\left\{ \frac{4\bar\theta}{\underline\theta}, \frac{r}{\alpha} \right\} \frac{\max\{\beta^2, 4\}}{b-a} \sum_{j=1}^{r'-1} \inf_{w\in\mathbb{R}} \int_{z_{j-1}}^{z_{j+1}} (g(t)-w)^2\,\mathrm{d}t
$$

$$
\le \max\left\{ \frac{4\bar\theta}{\underline\theta}, \frac{r}{\alpha} \right\} \frac{\max\{\beta^2, 4\}}{b-a} 2 \inf_{w\in\mathbb{R}} \int_a^b (g(t)-w)^2\,\mathrm{d}t \tag{89}
$$

$$
= \max\left\{ \frac{\bar\theta}{\underline\theta}, \frac{r}{4\alpha} \right\} \frac{\max\{8\beta^2, 32\}}{b-a} \inf_{w\in\mathbb{R}} \int_a^b (g(t)-w)^2\,\mathrm{d}t
$$

By (87), (88) and (89) we have

$$
\left( \int_a^b \sqrt{q(t)(1-q(t))}\,\mathrm{d}V_g([a,t]) \right)^2
$$

$$
\le 2r \max\left\{ \frac{\bar\theta}{\underline\theta}, \frac{r}{4\alpha} \right\} \frac{\max\{9\beta^2, 32+\beta^2\}}{b-a} \inf_{w\in\mathbb{R}} \int_a^b (g(t)-w)^2\,\mathrm{d}t
$$

Hence (78) is true, and the proof of Proposition 3.2 is complete.

## B.4 Proof of Example 3.1

Given $[a, b] \subseteq [0, 1]$, without loss of generality, assume $\int_a^b g(t) = 0$ (because the infimum in $w$ is achieved at $w = \int_a^b g(t)$). Let $t_0 \in [a, b]$ be the point with $g(t_0) = 0$. Since $g'(t) \geq c_1 > 0$, we have

$$\int_{t_0}^b (g(t))^2 \, dt \geq \int_{t_0}^b (c_1(t - t_0))^2 \, dt = \frac{c_1^2}{3}(b - t_0)^3$$

Similarly,

$$\int_a^{t_0} (g(t))^2 \, dt \geq \int_a^{t_0} (c_1(t_0 - t))^2 \, dt = \frac{c_1^2}{3}(t_0 - a)^3$$

As a result, we have

$$\int_a^b (g(t))^2 \, dt \geq \frac{c_1^2}{3}\left((b - t_0)^3 + (t_0 - a)^3\right) \geq \frac{2c_1^2}{3}\left(\frac{b - a}{2}\right)^3 = \frac{c_1^2}{12}(b - a)^3 \tag{90}$$

On the other hand, since $|g'(t)| \leq c_2$, we have

$$\left(\int_a^b |g'(t)| \, dt\right)^2 \leq c_2^2(b - a)^2 \tag{91}$$

Combining (90) and (91), we have

$$\left(\int_a^b |g'(t)| \, dt\right)^2 \leq \frac{12c_2^2}{c_1^2(b - a)} \int_a^b (g(t))^2 \, dt$$

## B.5 Proof of Example 3.3

It suffices to prove that there exists a constant $C_r$ such that for any univariate polynomial with a degree at most $r$ and for any $a < b$,

$$\left(\int_a^b |g'(t)| \, dt\right)^2 \leq \frac{C_r}{b - a} \int_a^b |g(t)|^2 \, dt \tag{92}$$

We first prove the conclusion when $a = 0$ and $b = 1$. Let $\mathcal{P}(r)$ be the set of all univariate polynomials with degree at most $r$. Note that $\mathcal{P}(r)$ is a finite-dimensional linear space, and the differential operator $\Phi : g \mapsto g'$ is a linear mapping on $\mathcal{P}(r)$. As a result, there exists $C_r$ such that

$$\int_0^1 |g'(t)| \, dt \leq \sqrt{C_r} \int_0^1 |g(t)| \, dt$$

for all $g \in \mathcal{P}(r)$.

For general $a < b$, given $g \in \mathcal{P}(r)$, define $h(s) := g(a + (b - a)s)$, then $h \in \mathcal{P}(r)$. So we have

$$\int_0^1 |h'(s)| \, ds \leq \sqrt{C_r} \int_0^1 |h(s)| \, ds \tag{93}$$

Note that

$$\int_0^1 |h'(s)| \, ds = (b - a) \int_0^1 |g'(a + (b - a)s)| \, ds = \int_a^b |g'(t)| \, dt \tag{94}$$

and

$$\int_0^1 |h(s)| \, ds = \int_0^1 |g(a + (b - a)s)| \, ds = \frac{1}{b - a} \int_a^b |g(t)| \, dt \tag{95}$$

Combining (93), (94) and (95), we know that

$$\left(\int_a^b |g'(t)| \, dt\right)^2 \leq \left(\frac{\sqrt{C_r}}{b - a} \int_a^b |g(t)| \, dt\right)^2 \leq \frac{C_r}{b - a} \int_a^b |g(t)|^2 \, dt$$

where the last step is by Cauchy-Schwarz inequality.

## B.6 Proof of Example 3.2

It suffices to prove that for any $a, b \in [0, 1]$ with $a < b$, it holds

$$\int_a^b |g'(t)| \, dt \leq \frac{110(L/\sigma)}{b - a} \int_a^b |g(t)| \, dt \tag{96}$$

Once (96) is proved, the conclusion is true via Jensen's inequality.

Below we prove (96). Denote $C = L/10$. For given $a, b \in [0, 1]$, without loss of generality, we assume the median of $g$ on $[a, b]$ is 0, i.e., $\int_a^b \mathbb{1}_{\{g(t) \geq 0\}} \, dt = \int_a^b \mathbb{1}_{\{g(t) < 0\}} \, dt = (b - a)/2$ (otherwise translate by a constant). We discuss two different cases. We denote $m := \min_{t \in [a,b]} \{|g'(t)|\}$ and $M := \max_{t \in [a,b]} \{|g'(t)|\}$.

(Case 1) $m \geq C(b - a)$. Since $g$ is $L$-smooth on $[0, 1]$, we have

$$M \leq m + L(b - a)$$

Hence we have

$$\frac{M}{m} \leq 1 + \frac{L(b - a)}{m} \leq 1 + L/C \tag{97}$$

where the second inequality is by the assumption of (Case 1). Since $\min_{t \in [a,b]} \{|g'(t)|\} = m > 0$, without loss of generality, we assume that $g'(t) > 0$ for all $t \in [a, b]$. Denote $t_0 = (a + b)/2$. By our assumption that the median of $g$ on $[a, b]$ is 0, we know that $g(t_0) = 0$. Since $g$ is convex, for any $t \in [a, t_0]$, we have

$$g(t_0) - g(t) \geq \frac{t_0 - t}{t_0 - a}(g(t_0) - g(a))$$

which implies $g(t) \leq \frac{t_0 - t}{t_0 - a}(g(a) - g(t_0)) \leq 0$. As a result, we have

$$\int_a^{t_0} |g(t)| \, dt \geq |g(t_0) - g(a)|\frac{t_0 - a}{2} \geq \frac{(t_0 - a)^2}{2}m = \frac{(b - a)^2}{8}m \tag{98}$$

On the other hand,

$$\int_a^b |g'(t)| \, dt \leq M(b - a) \tag{99}$$

Combining (98) and (99) we have

$$(b - a)\int_a^b |g'(t)| \, dt \leq \frac{8M}{m}\int_a^b |g(t)| \, dt \leq 8(1 + L/C)\int_a^b |g(t)| \, dt = 88\int_a^b |g(t)| \, dt$$

where the second inequality made use of (97).

(Case 2) $m < C(b - a)$. Then by the $L$-smoothness of $g$ we have

$$M \leq (C + L)(b - a)$$

so we have

$$\int_a^b |g'(t)| \, dt \leq M(b - a) \leq (C + L)(b - a)^2 \tag{100}$$

Define interval $[t_1, t_2] := \{t \in [a, b] \mid g(t) \leq 0\}$. By our assumption that the median of $g$ on $[0, 1]$ is 0, we have $t_2 - t_1 = (b - a)/2$. Denote $t_0 = \arg\min_{t \in [a,b]} g(t)$. Define function $f$ on $[t_1, t_2]$:

$$f(t) := \begin{cases} g(t_0) \cdot (t - t_1)/(t_0 - t_1) & t \in [t_1, t_0] \\ g(t_0) \cdot (t_2 - t)/(t_2 - t_0) & t \in [t_0, t_2] \end{cases}$$

Then $0 \geq f(t) \geq g(t)$ for all $t \in [t_1, t_2]$ (because $g$ is convex). Note that

$$\int_{t_1}^{t_2} f(t) \, dt = \frac{1}{2}g(t_0)(t_0 - t_1) + \frac{1}{2}g(t_0)(t_2 - t_0) = \frac{1}{2}g(t_0)(t_2 - t_1) = \frac{1}{2}\int_{t_1}^{t_2} g(t_0) \, dt \tag{101}$$

As a result,

$$\int_{t_1}^{t_2} |g(t)| \, dt \geq \int_{t_1}^{t_2} |f(t)| \, dt = -\int_{t_1}^{t_2} f(t) \, dt = \int_{t_1}^{t_2} f(t) - g(t_0) \, dt \geq \int_{t_1}^{t_2} g(t) - g(t_0) \, dt \tag{102}$$

where the first and last inequalities are because $0 \geq f(t) \geq g(t)$ for all $t \in [t_1, t_2]$; the second equality is by (101). Note that for any $t \in [t_1, t_2]$,

$$g(t) - g(t_0) \geq g'(t_0)(t - t_0) + \frac{\sigma}{2}(t - t_0)^2 \geq \frac{\sigma}{2}(t - t_0)^2 \tag{103}$$

where the first inequality is because $g$ is $\sigma$-strongly-convex, and the second is because $t_0$ is the minimizer of $g$ on $[t_1, t_2]$. By (102) and (103), we have

$$\int_{t_1}^{t_2} |g(t)| \, dt \geq \frac{\sigma}{2}\int_{t_1}^{t_2} (t - t_0)^2 \, dt \geq 2 \cdot \frac{\sigma}{2}\int_0^{(t_2 - t_1)/2} s^2 \, dt = \frac{\sigma}{24}(t_2 - t_1)^3 = \frac{\sigma}{192}(b - a)^3$$

Since $g$ is convex on $[a, b]$, the median of $g$ on $[a, b]$ is 0, and $[t_1, t_2] = \{t \in [a, b] \mid g(t) \leq 0\}$, it is not hard to check that

$$\int_a^b |g(t)| \, dt \geq 2\int_{t_1}^{t_2} |g(t)| \, dt \geq \frac{\sigma}{96}(b - a)^3 \tag{104}$$

Combining (100) and (104) we have

$$\int_a^b |g'(t)| \, dt \leq \frac{1}{b - a} \cdot \frac{96(C + L)}{\sigma}\int_a^b |g(t)| \, dt \leq \frac{110(L/\sigma)}{b - a}\int_a^b |g(t)| \, dt$$

where the last inequality made use of $C = L/10$.

The proof is complete by combining the discussions in (Case 1) and (Case 2).

# C  Comparison of Theorem 2.3 and Theorem 1 of [10]

We first restate Theorem 1 of [10] in the setting of fitting a single tree (note that [10] discussed random forest).

**Proposition C.1** *(Theorem 1 of [10]) Suppose Assumptions 2.2 and 2.1 hold true. Let $\widehat{f}^{(d)}(\cdot)$ be the tree estimated by CART with depth $d$. Fixed constants $\alpha_2 > 1$, $0 < \eta < 1/8$, $0 < c < 1/4$ and $\delta > 0$ with $2\eta < \delta < 1/4$. Then there exists constant $C > 0$ such that for all $n$ and $d$ satisfying $1 \le d \le c\log_2(n)$, it holds*

$$\mathbb{E}(\|\widehat{f}^{(d)} - f^*\|^2_{L^2(\mu)}) \le C\left(n^{-\eta} + (1 - \alpha_2^{-1}\lambda)^d + n^{-\delta+c}\right) \tag{105}$$

*In particular, the RHS of* (105) *is lower bounded by*

$$\Omega(n^{-\eta} + n^{-\delta+c} + n^{c\log_2(1-\lambda)}) \tag{106}$$

Note that (106) follows (105) by the fact $1 \le d \le c\log_2(n)$ and $\alpha_2 > 1$. In the original Assumptions of Theorem 1 in [10], it was assumed that the noises can be heavy-tailed, which is a weaker assumption than Assumption 2.1. However, the parameter controlling the tails of the noises did not explicitly enter the error bound (105), and it seems that their proof techniques cannot improve the error bound even under the assumption that noises are bounded. In addition, the dependence on $p$ was not explicitly stated in the bound (105), which seems to be hidden in the constant $C$.

To compare our error bound with the error bound in (106), since the $\|\widehat{f}^{(d)} - f^*\|^2_{L^2(\mu)}$ is bounded almost surely, it is not hard to transform the high-probability bound in (11) to an bound in expectation, and we have

$$\mathbb{E}(\|\widehat{f}^{(d)} - f^*\|^2_{L^2(\mu)}) \le O(n^{-\phi(\lambda)}\log(np)\log^2(n)) \tag{107}$$

Below we discuss two different cases.

- (Case 1) $\lambda \ge 1/2$. Then it holds

$$\phi(\lambda) = \frac{-\log_2(1-\lambda)}{1 - \log_2(1-\lambda)} \ge \frac{-\log_2(1/2)}{1 - \log_2(1/2)} = 1/2 \tag{108}$$

  So our convergence rate in (107) is $O(n^{-1/2}\log(np)\log^2(n))$, but the rate in (106) is

$$\Omega(n^{-\eta} + n^{-\delta+c} + n^{c\log_2(1-\lambda)}) \ge \Omega(n^{-\eta}) \ge \Omega(n^{-1/8}) \tag{109}$$

- (Case 2) $0 < \lambda \le 1/2$. Then it holds

$$1 - \log_2(1-\lambda) \le 1 - \log_2(1/2) = 2 \tag{110}$$

  and hence $\phi(\lambda) \ge -\log_2(1-\lambda)/2$. So our rate in (107) is $O(n^{\log_2(1-\lambda)/2}\log(np)\log^2(n))$, but the rate in (106) is

$$\Omega(n^{-\eta} + n^{-\delta+c} + n^{c\log_2(1-\lambda)}) \ge \Omega(n^{c\log_2(1-\lambda)}) \ge \Omega(n^{\frac{1}{4}\log_2(1-\lambda)}) \tag{111}$$

# D  Auxiliary results

**Lemma D.1** *(Bernstein's inequality) Let $Z_1, ...., Z_n$ be i.i.d. random variables satisfying $|\mathbb{E}((Z_1 - \mathbb{E}(Z_1))^k)| \le (1/2)k!\gamma^2 b^{k-2}$ for some constants $\gamma, b > 0$ and for all $k \ge 2$. Then for any $t > 0$,*

$$\mathbb{P}\left(\left|\frac{1}{n}\sum_{i=1}^{n} Z_i - \mathbb{E}(Z_i)\right| > t\right) \le 2\exp\left(-\frac{n}{4}\left(\frac{t^2}{\gamma^2} \wedge \frac{t}{b}\right)\right)$$

**Lemma D.2** *(Binomial tail bound) Let $Z_1, ..., Z_n$ be i.i.d. random variables with $\mathbb{P}(Z_i = 1) = \alpha$ and $\mathbb{P}(Z_i = 0) = 1 - \alpha$. Then for any $t \in (0, 1)$,*

$$\mathbb{P}\left(\frac{1}{n}\sum_{i=1}^{n} Z_i > t\right) \le \exp\left(-n\left[t\log\left(\frac{t}{\alpha}\right) + (1-t)\log\left(\frac{1-t}{1-\alpha}\right)\right]\right)$$

**Lemma D.3** *For any $t \in (0, 3/4)$, $\log(1-t) > -t - t^2$.*

*Proof.* For $t \in (0, 3/4)$,

$$\log(1-t) + t + t^2 = \frac{t^2}{2} - \sum_{k=3}^{\infty}\frac{t^k}{k!} \ge \frac{t^2}{2} - \frac{1}{6}\sum_{k=3}^{\infty}t^k = \frac{t^2}{2} - \frac{t^3}{6(1-t)} > 0.$$

$\square$

**Lemma D.4** *Suppose $Z$ is a random variable satisfying $\mathbb{E}(e^{\lambda Z}) \leq e^{\lambda^2 \sigma^2/2}$ for all $\lambda \in \mathbb{R}$, where $\sigma > 0$ is a constant; then*

$$\mathbb{E}(|Z|^k) \leq 9\sigma^k k!$$

*Proof.* By Chernoff inequality it holds $\mathbb{P}(|Z| > t) \leq 2\exp(-t^2/(2\sigma^2))$ for all $t > 0$. As a result,

$$
\begin{aligned}
\mathbb{E}(|Z^k|/(k!\sigma^k)) \leq \mathbb{E}(e^{|Z|/\sigma}) &= \int_0^\infty e^t \mathbb{P}(|Z|/\sigma > t) \, dt \\
&\leq \int_0^\infty 2\exp\left(t - \frac{t^2}{2}\right) dt = 2\sqrt{e} \int_0^\infty \exp(-(t-1)^2/2) \, dt \\
&\leq 2\sqrt{e} \int_{-\infty}^\infty \exp(-t^2/2) \, dt = 2\sqrt{2\pi e} \leq 9
\end{aligned}
$$

$\square$

**Lemma D.5** *For any integer $k \geq 2$ it holds $\frac{1}{k^2} - \frac{4}{(k+1)^3} \leq \frac{1}{(k+1)^2}$.*

*Proof.* For any $k \geq 2$ it holds

$$\frac{(2k+1)(k+1)}{2k^2} = (1 + \frac{1}{2k})(1 + \frac{1}{k}) \leq (1 + \frac{1}{4})(1 + \frac{1}{2}) < 2$$

Multiplying $2/(k+1)^3$ in the display above, we have

$$\frac{2k+1}{k^2(k+1)^2} < \frac{4}{(k+1)^3}$$

The proof is complete by noting that $\frac{2k+1}{k^2(k+1)^2} = \frac{1}{k^2} - \frac{1}{(k+1)^2}$.

$\square$

**Lemma D.6** *Let $[a, b]$ be a sub-interval of $[0, 1]$, and $c \in (a, b)$. Let $h$ be a function on $[a, b]$ such that $h$ is differentiable on $(a, c)$ and $(c, b)$, but can be discontinuous at $c$. Denote $\Delta h(c) := \lim_{t \to c+} h(t) - \lim_{t \to c-} h(t)$. Suppose*

$$\Delta h(c) > 4\max\left\{\int_a^c |h'(t)| \, \mathrm{d}t, \ \int_c^b |h'(t)| \, \mathrm{d}t\right\} \tag{112}$$

*Then it holds*

$$\inf_{w \in \mathbb{R}} \int_a^b (h(t) - w)^2 \, \mathrm{d}t \geq \min\{c - a, b - c\}(\Delta h(c))^2/16$$

*Proof.* We assume that $h$ is not continuous at $c$, since otherwise, the conclusion holds true trivially. We use the notation $h(c+) := \lim_{t \to c+} h(t)$ and $h(c-) := \lim_{t \to c-} h(t)$. Without loss of generality, assume $h(c+) > h(c-)$.

For $w \geq (1/2)(h(c+) + h(c-))$, it holds $w - h(c-) \geq (1/2)\Delta h(c)$. By (112), we know that for any $t \in (a, c)$,

$$|h(t) - h(c-)| \leq \int_a^c |h'(\tau)| \, \mathrm{d}\tau \leq \frac{1}{4}\Delta h(c)$$

Hence for all $t \in (a, c)$,

$$w - h(t) = w - h(c-) + h(c-) - h(t) \geq \frac{1}{2}\Delta h(c) - \frac{1}{4}\Delta h(c) = \frac{1}{4}\Delta h(c)$$

As a result,

$$\int_a^b (h(t) - w)^2 \, \mathrm{d}t \geq \int_a^c (h(t) - w)^2 \, \mathrm{d}t \geq (c - a)(\Delta h(c))^2/16 \tag{113}$$

For $w < (1/2)(h(c+) + h(c-))$, similarly, we can prove

$$\int_a^b (h(t) - w)^2 \, \mathrm{d}t \geq (b - c)(\Delta h(c))^2/16 \tag{114}$$

The proof is complete by combining (113) and (114).

$\square$

