668    Note that

$$\int_0^1 |h'(s)| \, \mathrm{d}s = (b - a) \int_0^1 |g'(a + (b - a)s)| \, \mathrm{d}s = \int_a^b |g'(t)| \, \mathrm{d}t \tag{94}$$

669    and

$$\int_0^1 |h(s)| \, \mathrm{d}s = \int_0^1 |g(a + (b - a)s)| \, \mathrm{d}s = \frac{1}{b - a} \int_a^b |g(t)| \, \mathrm{d}t \tag{95}$$

670    Combining (93), (94) and (95), we know that

$$\left(\int_a^b |g'(t)| \, \mathrm{d}t\right)^2 \leq \left(\frac{\sqrt{C_r}}{b - a} \int_a^b |g(t)| \, \mathrm{d}t\right)^2 \leq \frac{C_r}{b - a} \int_a^b |g(t)|^2 \, \mathrm{d}t$$

671    where the last step is by Cauchy-Schwarz inequality.

672    **B.6    Proof of Example 3.2**

673    It suffices to prove that for any $a, b \in [0, 1]$ with $a < b$, it holds

$$\int_a^b |g'(t)| \, \mathrm{d}t \leq \frac{110(L/\sigma)}{b - a} \int_a^b |g(t)| \, \mathrm{d}t \tag{96}$$

674    Once (96) is proved, the conclusion is true via Jensen's inequality.

Below we prove (96). Denote $C = L/10$. For given $a, b \in [0, 1]$, without loss of generality, we assume the median of $g$ on $[a, b]$ is 0, i.e., $\int_a^b 1_{\{g(t) \geq 0\}} \mathrm{d}t = \int_a^b 1_{\{g(t) < 0\}} \mathrm{d}t = (b-a)/2$ (otherwise translate by a constant). We discuss two different cases. We denote $m := \min_{t \in [a,b]} \{|g'(t)|\}$ and $M := \max_{t \in [a,b]} \{|g'(t)|\}$.

(Case 1) $m \geq C(b - a)$. Since $g$ is $L$-smooth on $[0, 1]$, we have

$$M \leq m + L(b - a)$$

Hence we have

$$\frac{M}{m} \leq 1 + \frac{L(b-a)}{m} \leq 1 + L/C \tag{97}$$

where the second inequality is by the assumption of (Case 1). Since $\min_{t \in [a,b]} \{|g'(t)|\} = m > 0$, without loss of generality, we assume that $g'(t) > 0$ for all $t \in [a, b]$. Denote $t_0 = (a + b)/2$. By our assumption that the median of $g$ on $[a, b]$ is 0, we know that $g(t_0) = 0$. Since $g$ is convex, for any $t \in [a, t_0]$, we have

$$g(t_0) - g(t) \geq \frac{t_0 - t}{t_0 - a}(g(t_0) - g(a))$$

which implies $g(t) \leq \frac{t_0 - t}{t_0 - a}(g(a) - g(t_0)) \leq 0$. As a result, we have

$$\int_a^{t_0} |g(t)| \, \mathrm{d}t \geq |g(t_0) - g(a)| \frac{t_0 - a}{2} \geq \frac{(t_0 - a)^2}{2} m = \frac{(b - a)^2}{8} m \tag{98}$$

On the other hand,

$$\int_a^b |g'(t)| \, \mathrm{d}t \leq M(b - a) \tag{99}$$

Combining (98) and (99) we have

$$(b - a) \int_a^b |g'(t)| \, \mathrm{d}t \leq \frac{8M}{m} \int_a^b |g(t)| \, \mathrm{d}t \leq 8(1 + L/C) \int_a^b |g(t)| \, \mathrm{d}t = 88 \int_a^b |g(t)| \, \mathrm{d}t$$

where the second inequality made use of (97).

(Case 2) $m < C(b - a)$. Then by the $L$-smoothness of $g$ we have

$$M \leq (C + L)(b - a)$$

so we have

$$\int_a^b |g'(t)| \, \mathrm{d}t \leq M(b - a) \leq (C + L)(b - a)^2 \tag{100}$$

Define interval $[t_1, t_2] := \{t \in [a, b] \mid g(t) \leq 0\}$. By our assumption that the median of $g$ on $[0, 1]$ is 0, we have $t_2 - t_1 = (b - a)/2$. Denote $t_0 = \operatorname{argmin}_{t \in [a,b]} g(t)$. Define function $f$ on $[t_1, t_2]$:

$$f(t) := \begin{cases} g(t_0) \cdot (t - t_1)/(t_0 - t_1) & t \in [t_1, t_0] \\ g(t_0) \cdot (t_2 - t)/(t_2 - t_0) & t \in [t_0, t_2] \end{cases}$$

Then $0 \geq f(t) \geq g(t)$ for all $t \in [t_1, t_2]$ (because $g$ is convex). Note that

$$\int_{t_1}^{t_2} f(t) \, \mathrm{d}t = \frac{1}{2}g(t_0)(t_0 - t_1) + \frac{1}{2}g(t_0)(t_2 - t_0) = \frac{1}{2}g(t_0)(t_2 - t_1) = \frac{1}{2}\int_{t_1}^{t_2} g(t_0) \, \mathrm{d}t \tag{101}$$

As a result,

$$\int_{t_1}^{t_2} |g(t)| \, \mathrm{d}t \geq \int_{t_1}^{t_2} |f(t)| \, \mathrm{d}t = -\int_{t_1}^{t_2} f(t) \, \mathrm{d}t = \int_{t_1}^{t_2} f(t) - g(t_0) \, \mathrm{d}t \geq \int_{t_1}^{t_2} g(t) - g(t_0) \, \mathrm{d}t \tag{102}$$

where the first and last inequalities are because $0 \geq f(t) \geq g(t)$ for all $t \in [t_1, t_2]$; the second equality is by (101). Note that for any $t \in [t_1, t_2]$,

$$g(t) - g(t_0) \geq g'(t_0)(t - t_0) + \frac{\sigma}{2}(t - t_0)^2 \geq \frac{\sigma}{2}(t - t_0)^2 \tag{103}$$

where the first inequality is because $g$ is $\sigma$-strongly-convex, and the second is because $t_0$ is the minimizer of $g$ on $[t_1, t_2]$. By (102) and (103), we have

$$\int_{t_1}^{t_2} |g(t)| \, \mathrm{d}t \geq \frac{\sigma}{2} \int_{t_1}^{t_2} (t - t_0)^2 \, \mathrm{d}t \geq 2 \cdot \frac{\sigma}{2} \int_0^{(t_2 - t_1)/2} s^2 \, \mathrm{d}t = \frac{\sigma}{24}(t_2 - t_1)^3 = \frac{\sigma}{192}(b - a)^3$$

Since $g$ is convex on $[a, b]$, the median of $g$ on $[a, b]$ is 0, and $[t_1, t_2] = \{t \in [a, b] \mid g(t) \leq 0\}$, it is not hard to check that

$$\int_a^b |g(t)| \, \mathrm{d}t \geq 2 \int_{t_1}^{t_2} |g(t)| \, \mathrm{d}t \geq \frac{\sigma}{96}(b - a)^3 \tag{104}$$

Combining (100) and (104) we have

$$\int_a^b |g'(t)| \, \mathrm{d}t \leq \frac{1}{b - a} \cdot \frac{96(C + L)}{\sigma} \int_a^b |g(t)| \, \mathrm{d}t \leq \frac{110(L/\sigma)}{b - a} \int_a^b |g(t)| \, \mathrm{d}t$$

where the last inequality made use of $C = L/10$.

The proof is complete by combining the discussions in (Case 1) and (Case 2).

## C    Comparison of Theorem 2.3 and Theorem 1 of [10]

We first restate Theorem 1 of [10] in the setting of fitting a single tree (note that [10] discussed random forest).

**Proposition C.1** *(Theorem 1 of [10]) Suppose Assumptions 2.2, 2.1 and* ?? *hold true. Let $\widehat{f}^{(d)}(\cdot)$ be the tree estimated by CART with depth $d$. Fixed constants $\alpha_2 > 1$, $0 < \eta < 1/8$, $0 < c < 1/4$ and $\delta > 0$ with $2\eta < \delta < 1/4$. Then there exists constant $C > 0$ such that for all $n$ and $d$ satisfying $1 \leq d \leq c \log_2(n)$, it holds*

$$\mathbb{E}(\|\widehat{f}^{(d)} - f^*\|_{L^2(\mu)}^2) \leq C\left(n^{-\eta} + (1 - \alpha_2^{-1}\lambda)^d + n^{-\delta+c}\right) \tag{105}$$

*In particular, the RHS of* (105) *is lower bounded by*

$$\Omega(n^{-\eta} + n^{-\delta+c} + n^{c\log_2(1-\lambda)}) \tag{106}$$

Note that (106) follows (105) by the fact $1 \leq d \leq c \log_2(n)$ and $\alpha_2 > 1$. In the original Assumptions of Theorem 1 in [10], it was assumed that the noises can be heavy-tailed, which is a weaker assumption than Assumption 2.1. However, the parameter controlling the tails of the noises did not explicitly enter the error bound (105), and it seems that their proof techniques cannot improve the error bound even under the assumption that noises are bounded. In addition, the dependence on $p$ was not explicitly stated in the bound (105), which seems to be hidden in the constant $C$.

To compare our error bound with the error bound in (106), since the $\|\widehat{f}^{(d)} - f^*\|_{L^2(\mu)}^2$ is bounded almost surely, it is not hard to transform the high-probability bound in (11) to an bound in expectation, and we have

$$\mathbb{E}(\|\widehat{f}^{(d)} - f^*\|_{L^2(\mu)}^2) \leq O(n^{-\phi(\lambda)}\log(np)\log^2(n)) \tag{107}$$

Below we discuss two different cases.

- (Case 1) $\lambda \geq 1/2$. Then it holds

$$\phi(\lambda) = \frac{-\log_2(1-\lambda)}{1 - \log_2(1-\lambda)} \geq \frac{-\log_2(1/2)}{1 - \log_2(1/2)} = 1/2 \tag{108}$$

  So our convergence rate in (107) is $O(n^{-1/2}\log(np)\log^2(n))$, but the rate in (106) is

$$\Omega(n^{-\eta} + n^{-\delta+c} + n^{c\log_2(1-\lambda)}) \geq \Omega(n^{-\eta}) \geq \Omega(n^{-1/8}) \tag{109}$$

- (Case 2) $0 < \lambda \leq 1/2$. Then it holds

$$1 - \log_2(1-\lambda) \leq 1 - \log_2(1/2) = 2 \tag{110}$$

  and hence $\phi(\lambda) \geq -\log_2(1-\lambda)/2$. So our rate in (107) is $O(n^{\log_2(1-\lambda)/2}\log(np)\log^2(n))$, but the rate in (106) is

$$\Omega(n^{-\eta} + n^{-\delta+c} + n^{c\log_2(1-\lambda)}) \geq \Omega(n^{c\log_2(1-\lambda)}) \geq \Omega(n^{\frac{1}{4}\log_2(1-\lambda)}) \tag{111}$$

## D    Auxiliary results

**Lemma D.1** *(Bernstein's inequality) Let $Z_1, ...., Z_n$ be i.i.d. random variables satisfying $|\mathbb{E}((Z_1 - \mathbb{E}(Z_1))^k)| \leq (1/2)k!\gamma^2 b^{k-2}$ for some constants $\gamma, b > 0$ and for all $k \geq 2$. Then for any $t > 0$,*

$$\mathbb{P}\left(\left|\frac{1}{n}\sum_{i=1}^n Z_i - \mathbb{E}(Z_i)\right| > t\right) \leq 2\exp\left(-\frac{n}{4}\left(\frac{t^2}{\gamma^2} \wedge \frac{t}{b}\right)\right)$$

**Lemma D.2** *(Binomial tail bound) Let $Z_1, ..., Z_n$ be i.i.d. random variables with $\mathbb{P}(Z_i = 1) = \alpha$ and $\mathbb{P}(Z_i = 0) = 1 - \alpha$. Then for any $t \in (0, 1)$,*

$$\mathbb{P}\left(\frac{1}{n}\sum_{i=1}^n Z_i > t\right) \leq \exp\left(-n\left[t\log\left(\frac{t}{\alpha}\right) + (1-t)\log\left(\frac{1-t}{1-\alpha}\right)\right]\right)$$

**Lemma D.3** *For any $t \in (0, 3/4)$, $\log(1-t) > -t - t^2$.*

*Proof.* For $t \in (0, 3/4)$,

$$\log(1-t) + t + t^2 = \frac{t^2}{2} - \sum_{k=3}^{\infty}\frac{t^k}{k!} \geq \frac{t^2}{2} - \frac{1}{6}\sum_{k=3}^{\infty}t^k = \frac{t^2}{2} - \frac{t^3}{6(1-t)} > 0.$$

$\square$

**Lemma D.4** *Suppose $Z$ is a random variable satisfying $\mathbb{E}(e^{\lambda Z}) \leq e^{\lambda^2 \sigma^2/2}$ for all $\lambda \in \mathbb{R}$, where $\sigma > 0$ is a constant; then*

$$\mathbb{E}(|Z|^k) \leq 9\sigma^k k!$$

*Proof.* By Chernoff inequality it holds $\mathbb{P}(|Z| > t) \leq 2\exp(-t^2/(2\sigma^2))$ for all $t > 0$. As a result,

$$
\begin{aligned}
\mathbb{E}(|Z^k|/(k!\sigma^k)) \leq \mathbb{E}(e^{|Z|/\sigma}) &= \int_0^\infty e^t \mathbb{P}(|Z|/\sigma > t)\, dt \\
&\leq \int_0^\infty 2\exp\left(t - \frac{t^2}{2}\right) dt \;=\; 2\sqrt{e}\int_0^\infty \exp(-(t-1)^2/2)\, dt \\
&\leq 2\sqrt{e}\int_{-\infty}^\infty \exp(-t^2/2)\, dt \;=\; 2\sqrt{2\pi e} \leq 9
\end{aligned}
$$

$\square$

**Lemma D.5** *For any integer $k \geq 2$ it holds $\frac{1}{k^2} - \frac{4}{(k+1)^3} \leq \frac{1}{(k+1)^2}$.*

*Proof.* For any $k \geq 2$ it holds

$$\frac{(2k+1)(k+1)}{2k^2} = (1 + \frac{1}{2k})(1 + \frac{1}{k}) \leq (1 + \frac{1}{4})(1 + \frac{1}{2}) < 2$$

Multiplying $2/(k+1)^3$ in the display above, we have

$$\frac{2k+1}{k^2(k+1)^2} < \frac{4}{(k+1)^3}$$

The proof is complete by noting that $\frac{2k+1}{k^2(k+1)^2} = \frac{1}{k^2} - \frac{1}{(k+1)^2}$.

$\square$

**Lemma D.6** *Let $[a,b]$ be a sub-interval of $[0,1]$, and $c \in (a,b)$. Let $h$ be a function on $[a,b]$ such that $h$ is differentiable on $(a,c)$ and $(c,b)$, but can be discontinuous at $c$. Denote $\Delta h(c) := \lim_{t\to c+} h(t) - \lim_{t\to c-} h(t)$. Suppose*

$$\Delta h(c) > 4\max\left\{\int_a^c |h'(t)|\, dt,\ \int_c^b |h'(t)|\, dt\right\} \tag{112}$$

*Then it holds*

$$\inf_{w\in\mathbb{R}} \int_a^b (h(t) - w)^2\, dt \;\geq\; \min\{c-a, b-c\}(\Delta h(c))^2/16$$

*Proof.* We assume that $h$ is not continuous at $c$, since otherwise, the conclusion holds true trivially. We use the notation $h(c+) := \lim_{t\to c+} h(t)$ and $h(c-) := \lim_{t\to c-} h(t)$. Without loss of generality, assume $h(c+) > h(c-)$.

For $w \geq (1/2)(h(c+) + h(c-))$, it holds $w - h(c-) \geq (1/2)\Delta h(c)$. By (112), we know that for any $t \in (a,c)$,

$$|h(t) - h(c-)| \leq \int_a^c |h'(\tau)|\, d\tau \leq \frac{1}{4}\Delta h(c)$$

Hence for all $t \in (a,c)$,

$$w - h(t) = w - h(c-) + h(c-) - h(t) \geq \frac{1}{2}\Delta h(c) - \frac{1}{4}\Delta h(c) = \frac{1}{4}\Delta h(c)$$

As a result,

$$\int_a^b (h(t) - w)^2\, dt \geq \int_a^c (h(t) - w)^2\, dt \geq (c-a)(\Delta h(c))^2/16 \tag{113}$$

For $w < (1/2)(h(c+) + h(c-))$, similarly, we can prove

$$\int_a^b (h(t) - w)^2\, dt \geq (b-c)(\Delta h(c))^2/16 \tag{114}$$

The proof is complete by combining (113) and (114).

$\square$