# OpenReview forum: "On the Convergence of CART under Sufficient Impurity Decrease Condition"
_NeurIPS.cc/2023/Conference — NeurIPS 2023 poster_

### Official Review · Reviewer_te2H · 2023-07-03

**Soundness:** 4 excellent
**Presentation:** 4 excellent
**Contribution:** 3 good
**Rating:** 6
**Confidence:** 3

**Summary:**

This paper improved the rate of consistency of CART based on a sufficient impurity decrease (SID) condition under regression settings. Then, the authors provided examples, which are mostly special additive models, that can satisfy the SID condition, and showed that the rate of consistency cannot be improved by more than a log factor.

**Strengths:**

The strengths of this paper are summarized as follows:

- The "locally reverse Poincare inequality" shown in this paper is intuitive and can be more general than previous studies.
- The results are intuitive and improved the results in Chi et al. 2022 when the noise is sub-Gaussian.

**Weaknesses:**

The weaknesses of this paper are summarized as follows:

**About contributions**

1. The examples in this paper that satisfy the SID condition are still restricted to special additive models, which can be somehow not so practical. And it seems that extending the results to non-additive models is not easy.
2. The discussions about optimality in [Line 279-285] are not so strict. Note that the lower bound of order $\Omega(n^{-2 / (p + 2)})$ shown in Tan et al. 2022 is based on general additive models which may not satisfy the SID condition. In other words, when the SID condition satisfies, the lower bound may be better.

Typos:
- $\kappa$ in [Line 242] seems to be not defined and it seems to be $\tau$.

**Questions:**

I would like to ask the authors the following questions:

- Can the SID condition be satisfied in multi-dimensional settings without assuming additive models? See Weaknesses #1.
- May the curse of dimensionality be avoided when the SID condition verifies? See Weaknesses #2.

**Limitations:**

The authors adequately addressed the limitations.

---

> ### Author Rebuttal · Authors · 2023-08-08
>
> Thank you for your overall positive assessment of our work.
>
>
> Reply to weakness 1/ question 1:
>
> Thank you for this comment. Actually, it is possible to relax the additive model on $f^*$ such that it is ``approximately additive". More precisely, we can assume that there is an additive function $g^*$ that approximates $f^*$, such that in each rectangle A, the L2 distance between $f^*$ and $g^*$ is bounded by a small constant of the variance of $f^*$ on $A$.
>
> On the other hand, it seems hard to further relax this condition. Consider a two dimensional model $f^*(x_1,x_2)$ with features $X_1$ and $X_2$ independent. Suppose $E(f^*(X_1,X_2)) = 0$.
>     We can decompose $f^*$ into additive components and a remaining term:
>     \begin{equation}
>     f^*(x_1,x_2) = f_1^*(x_1) + f_2^*(x_2) + h^*(x_1,x_2)
>     \end{equation}
>
>  where $f^*_1(x_1) := E(f^*(X_1,X_2) | X_1=x_1)$, $f^*_2(x_2) := E(f^*(X_1,X_2) | X_2=x_2)$ and $h^*(x_1,x_2) := f^*(x_1,x_2) - f_1^*(x_1) - f_2^*(x_2)$. Then it can be checked that
>     \begin{equation}
>     E (h^*(X_1,X_2) | X_1) = E (h^*(X_1,X_2) | X_2) = 0
>     \end{equation}
>
> If the additive components $f^*_1 + f^*_2$ are small, say, in the extreme case $f^*_1 + f^*_2$ is zero, then $f^* = h^*$, and by the inequality above we know that the SID condition cannot be satisfied. Therefore, for the SID condition to be satisfied, it is necessary that the function $f^*$ has a significant amount of signal that is additive.
>
>  But of course, this is for the SID condition to be satisfied. It is possible that CART can still work well without the SID condition,
>     which needs more exploration and is beyond the scope of this work.
>
>
>
> Reply to weakness 2/ question 2:
>
> Thank you for this comment. We will reword the discussion on optimality in line 279-285.
>     Indeed, if we consider the class of functions satisfying SID condition with a fixed parameter $\lambda$, then the rate of Theorem 2.3 has an exponent with no explicit dependence dimension $p$. However, in multi-dimensional models, even for additive models satisfying the SID condition, the SID parameter $\lambda$ seems to have a dependence on $p$. Therefore it is not proper to claim that it avoids the curse of dimensionality.

---

> > ### Comment · Reviewer_te2H · 2023-08-17
> >
> > Thank you for providing such a comprehensive clarification. However, the dependency on the additive structure remains unrealistic, so I will keep my score.

---

> > > ### Author Response · Authors · 2023-08-17
> > >
> > > Thank you for your reply. We admit that the additive structure may look a little unrealistic. But still, we'd like to note that the additive structure assumption has also appeared in many existing recent literature analyzing CART:
> > >
> > > 1. Erwan Scornet, Gérard Biau, and Jean-Philippe Vert. Consistency of random forests. The Annals of
> > > 385 Statistics, 43(4):1716–1741, 2015.
> > >
> > > 2. Klusowski J. Sparse learning with CART[J]. Advances in Neural Information Processing Systems, 2020, 33: 11612-11622.
> > >
> > > 3. Klusowski J, Tian P. Large-scale prediction with decision trees[J]. Journal of the American Statistical Association, 2022.
> > >
> > > No existing work has shown the consistency of CART without making strong structural assumptions on the underlying model.
> > > On the other hand, the requirement for additive model assumption sheds light on the limitation of only using axis-aligned splits. Indeed, if skewed splits are allowed, the assumption on the underlying model can be significantly relaxed:
> > >
> > > 4. Cattaneo M D, Chandak R, Klusowski J M. Convergence rates of oblique regression trees for flexible function libraries[J]. arXiv preprint arXiv:2210.14429, 2022.
> > >
> > > Given that our major focus is on an analysis of CART under SID condition, we take the additive model as an example (as it appears in the literature) and use it to illustrate the applicability of SID condition. Of course, it can be an interesting (and challenging) question to relax these conditions in the future.

---

### Official Review · Reviewer_xrLy · 2023-07-06

**Soundness:** 3 good
**Presentation:** 3 good
**Contribution:** 3 good
**Rating:** 6
**Confidence:** 4

**Summary:**

The paper studies the convergence rate of CART, a greedy algorithm for building decision trees, under a regression setting. It introduces a sufficient impurity decrease (SID) condition on the underlying function that ensures the consistency and polynomial convergence of CART. It also provides examples and sufficient conditions to verify the SID condition for various function classes, such as additive models, polynomials, and smooth and strongly convex functions.

**Strengths:**

1.	The paper provides a refined analysis of CART under the SID condition, which improves the prediction error bound over the previous work by Chi et al. (2022) and shows its optimality up to logarithmic factors.
2.	The paper decodes the mystery of the SID condition by introducing a locally reverse Poincare (LRP) class of univariate functions and showing that additive functions with LRP components satisfy the SID condition. This connects the two types of assumptions in the literature: additive model and SID condition.
3.	The paper demonstrates the practical utility of its results by providing examples and sufficient conditions to check the LRP property for various well-known function classes, such as polynomials, smooth and strongly convex functions, and strongly increasing functions.
4.	The paper is well-written and clear, with detailed proofs and explanations in the appendix. It also includes simulations and figures to illustrate its main findings and compare them with existing results.


**Weaknesses:**

1.	The paper relies on some technical assumptions and conditions, such as bounded errors, bounded signal function, and LRP property, which may not hold or be easy to verify for some function classes. It would be helpful to provide more intuition and motivation for these assumptions and conditions, and discuss their implications and limitations for the applicability of the results.
2.	The improvement of the convergence rate in this paper still heavily relies on the SID condition. Although the paper decomposes and illustrates the SID condition with some examples, they are still mathematical forms in 1-dimensional situations. If the paper could further analyze the SID condition in relation to the structure of real data, then its contribution to the SID condition would be more convincing.


**Questions:**

1.	Do Examples 3.1 - 3.3 still hold if they are generalized to high-dimensional situations?
2.	The additive model assumption in Section 3 is still quite strong, is it possible to use milder assumptions?
3.	The intuition behind the LRP condition needs further explanation, does this mathematical property have any implications for improving the CART algorithm? Maybe it can help explain why CART tends to handle data with certain structural characteristics.


**Limitations:**

NAN.

---

> ### Author Rebuttal · Authors · 2023-08-08
>
> Thank you for your overall positive assessment of our work.
>
> Reply to weakness 1:
>
> Thank you for this suggestion, we will add a discussion in the revised paper.
>
>
> Reply to weakness 2 and question 2:
>
> Thank you for this comment. Actually, it is possible to relax the additive model on $f^*$ such that it is ``approximately additive". More precisely, we can assume that there is an additive function $g^*$ that approximates $f^*$, such that in each rectangle A, the L2 distance between $f^*$ and $g^*$ is bounded by a small constant of the variance of $f^*$ on $A$.
>
> On the other hand, it seems hard to further relax this condition. Consider a two dimensional model $f^*(x_1,x_2)$ with features $X_1$ and $X_2$ independent. Suppose $E(f^*(X_1,X_2)) = 0$.
>     We can decompose $f^*$ into additive components and a remaining term:
>     \begin{equation}
>     f^*(x_1,x_2) = f_1^*(x_1) + f_2^*(x_2) + h^*(x_1,x_2)
>     \end{equation}
>
>  where $f^*_1(x_1) := E(f^*(X_1,X_2) | X_1=x_1)$, $f^*_2(x_2) := E(f^*(X_1,X_2) | X_2=x_2)$ and $h^*(x_1,x_2) := f^*(x_1,x_2) - f_1^*(x_1) - f_2^*(x_2)$. Then it can be checked that
>     \begin{equation}
>     E (h^*(X_1,X_2) | X_1) = E (h^*(X_1,X_2) | X_2) = 0
>     \end{equation}
>
> If the additive components $f^*_1 + f^*_2$ are small, say, in the extreme case $f^*_1 + f^*_2$ is zero, then $f^* = h^*$, and by the inequality above we know that the SID condition cannot be satisfied. Therefore, for the SID condition to be satisfied, it is necessary that the function $f^*$ has a significant amount of signal that is additive.
>
>  But of course, this is for the SID condition to be satisfied. It is possible that CART can still work well without the SID condition,
>     which needs more exploration and is beyond the scope of this work.
>
>
>
> Reply to question 1:
>
> Thanks -- these are great questions. We believe that Examples 3.1 and 3.2 still satisfy the SID condition in multi-dimensional settings (for Example 3.1 we assume strict monotonicity in all the dimensions).
> For Example 3.3 we are not sure, as multi-variate polynomials may have some symmetric structure (like the XOR gate) that makes a single axis-aligned split fail to work.
>
> Reply to question 3:
>
> Thanks for this helpful comment. LRP condition indicates that CART can handle the signal in which there is a significant local variability (characterized by the derivative). We will add some comments in the revised version.

---

> > ### Comment · Reviewer_xrLy · 2023-08-20
> >
> > Thank you for your detailed response. However, since the conditions on which the results of this paper rely still require more explanation, I will keep the current score. Nevertheless, I would still like this paper to be accepted by the conference.

---

### Official Review · Reviewer_MZ6v · 2023-07-06

**Soundness:** 3 good
**Presentation:** 3 good
**Contribution:** 3 good
**Rating:** 5
**Confidence:** 1

**Summary:**

The paper performs theoretical analysis of the well-known CART algorithm. The authors show a convergence rate of CART under the condition called 'sufficient impurity decrease' (SID), which is tighter than known ones. The authors further provide the condition for a class of functions that satisfies SID.

**Strengths:**

- The paper seems technically sound and the analysis of CART would be important.

- The paper elaborately explains differences from past work in particular from [10].

**Weaknesses:**

- Experimental verification is only in one quite simple figure for the simplest case. Of-course, the paper is a theoretical paper, but if richer empirical evidences had been provided, it would have been convincing more (such as different true functions and comparison with past bounds).

**Questions:**

- Is there any insight that can be derived by the proposed analysis for tree ensemble models such as random forest?

**Limitations:**

A limitation about lambda is discussed in Section 5.

---

> ### Author Rebuttal · Authors · 2023-08-08
>
> Thank you for your overall positive assessment of our work.
>
> Reply to weakness 1:
>
> Thank you for this suggestion. The reason that we only did the simulation for linear functions is that for other signal functions, it is hard to precisely evaluate the SID parameter $\lambda$, hence difficult to set up a fair baseline for numerical evaluation. We would like to do more explorations in this direction in future works.
>
> Reply to question 1:
>
> Thank you for this question -- it immediately leads to considerations for future works. By a similar argument as in [10], we can immediately derive an error bound for regression random forests. But on the other hand, this error bound is at essence for a single tree and does not make use of the ensemble structure. If the structure of ensembling is properly analyzed, it is possible to derive a stronger error bound than the one given in this paper.

---

### Official Review · Reviewer_vX7v · 2023-07-10

**Soundness:** 3 good
**Presentation:** 3 good
**Contribution:** 3 good
**Rating:** 6
**Confidence:** 3

**Summary:**

The paper focuses on the analysis of the prediction error of Classification and Regression Trees (CART) for regression problems under a sufficient impurity decrease (SID) condition. The SID condition is a strong assumption on the approximation power of tree splits, which can ensure the consistency of CART. The paper establishes an upper bound on the prediction error of CART under the SID condition and shows that the error bound is tight up to log factors. The paper also discusses a few sufficient conditions under which an additive model satisfies the SID condition. The paper's first contribution is a refined analysis of CART under the SID condition, which improves upon the known result by providing a tighter error bound. The paper's second contribution is the decoding of the mystery of the SID condition, which builds a connection between the two types of assumptions in the literature: the additive model and the SID condition. The paper discusses a few examples of how the locally reverse Poincare inequality can be verified. Overall, the paper provides a refined analysis of CART under the SID condition and sheds light on the connection between two types of assumptions in the literature: additive model and SID condition.

**Strengths:**

The paper has several strengths. First, it provides a refined analysis of the prediction error of Classification and Regression Trees (CART) under a sufficient impurity decrease (SID) condition for regression problems. The paper establishes an upper bound on the prediction error of CART under the SID condition and shows that the error bound is tight up to log factors. Second, the paper discusses a few sufficient conditions under which an additive model satisfies the SID condition. Third, the paper provides examples of how the locally reverse Poincare inequality can be verified.

**Weaknesses:**

I think the work is significant. I don’t have many concerns, but I have some questions about the method.
1. Can this method be extended to classification problems?
2. In theorem 2.3, why do you need to suppose n satisfies such a complicated condition?
3. Also, in theorem 2.3, why should d be fixed to a function of data size?
4. In line 32, is it [0,1]^p?


**Questions:**

Referred to Strengths and Weaknesses

**Limitations:**

Referred to Strengths and Weaknesses

---

> ### Author Rebuttal · Authors · 2023-08-08
>
> Thank you for your overall positive assessment of our work.
>
> Reply to weakness 1:
>
> Thanks for this great question. It depends on the specific version of classification trees under discussion and the loss used to make splits and measure the accuracy. If the prediction in each node is the majority votes and the loss is the 0-1 loss, then the current analysis does not apply, because the impurity decrease considered in this paper is specific for square loss.
> If the prediction in each node is the empirical probability of lying in a class, and the loss is continuous with curvature (e.g. cross-entropy), then it is possible to generalize some of the analysis for classification.
>
> Reply to weakness 2:
>
> Thanks for this question. The condition on n is needed for some uniform bound between the empirical and population means of some quantities. This condition is mild in this context:
> If $\bar{\theta}$ and $\delta$ are constants, then this condition only requires that n is large enough with $dlog(np)/n \le O(1)$.
>
> Reply to weakness 3:
>
> Thanks for this question.
> Note that the error bound in (10) holds true for an arbitrary value of d, while the error bound in (11) is true only when $d \sim log_2(n)$.
> We take $d \sim log_2(n)$ simply to minimize the RHS of (10) and
> arrive at the error bound in (11).
> This value increases as $n$ increases, because when more data is available, a larger model can be used to decrease the bias without significantly increasing the variance.
>
> Reply to weakness 4:
>
> Thank you. We will correct it in the revision.

---

### Official Review · Reviewer_1fUr · 2023-07-20

**Soundness:** 3 good
**Presentation:** 4 excellent
**Contribution:** 3 good
**Rating:** 6
**Confidence:** 2

**Summary:**

In this paper, the approximation of the model obtained from the decision tree learning algorithm CART is analyzed. More precisely, in Theorem 2.3, a convergence rate of the decision tree approximation compared to the true model is obtained. This analysis is performed under two assumptions of the paper (already introduced in the literature), with the Sufficient Impurity Decrease (SID) assumption in particular. Finally, examples of true models that satisfy the SID assumption are proven.

**Strengths:**

1. The paper is well-written and easy to follow (although I did not check the proofs).
2. The analysis of Theorem 2.3 improves the result (i.e., the convergence rate) of [10].
3. Proposition 3.1 links the SID condition and the class of functions satisfying the Locally Reverse Poincaré condition (that is proven for some classes).

**Weaknesses:**

1. The additive structure assumption of $f^*$ seems restrictive (but is also assumed in previous works).


**Questions:**


Typos and minor comments:
- l38: the the -> the
- l104: Chi et al.[10] -> Chi et al. [10]
- l144: the notations $\bar{y}\_{\mathcal{I}\_A}$, $\bar{y}\_{\mathcal{I}\_{A\_L}}$ and $\bar{y}\_{\mathcal{I}\_{A\_R}}$ are not introduced
- l163: "as" -> "at"
- l193: "Assumptions‘2.1" -> "Assumptions‘ 2.1"
- l240/246/247: Poincare -> Poincaré
- l242: $\kappa$ -> "$\tau$"
- l704: there is an indefinite reference "??"
- The references are not "standardized", e.g. for JMLR it is written "The Journal of Machine Learning Research" or "Journal of Machine Learning Research"

**Limitations:**

The limitation about the the SID condition is discussed in Section 5. Moreover, I do not see any potential negative societal impact of their work.

---

> ### Author Rebuttal · Authors · 2023-08-08
>
> Thank you for your overall positive assessment of our work.
>
>
> Reply to weakness 1:
>
> Thank you for this comment. Actually, it is possible to relax the additive model on $f^*$ such that it is ``approximately additive". More precisely, we can assume that there is an additive function $g^*$ that approximates $f^*$, such that in each rectangle A, the L2 distance between $f^*$ and $g^*$ is bounded by a small constant of the variance of $f^*$ on $A$.
>
> On the other hand, it seems hard to further relax this condition. Consider a two dimensional model $f^*(x_1,x_2)$ with features $X_1$ and $X_2$ independent. Suppose $E(f^*(X_1,X_2)) = 0$.
>     We can decompose $f^*$ into additive components and a remaining term:
>     \begin{equation}
>     f^*(x_1,x_2) = f_1^*(x_1) + f_2^*(x_2) + h^*(x_1,x_2)
>     \end{equation}
>
>  where $f^*_1(x_1) := E(f^*(X_1,X_2) | X_1=x_1)$, $f^*_2(x_2) := E(f^*(X_1,X_2) | X_2=x_2)$ and $h^*(x_1,x_2) := f^*(x_1,x_2) - f_1^*(x_1) - f_2^*(x_2)$. Then it can be checked that
>     \begin{equation}
>     E (h^*(X_1,X_2) | X_1) = E (h^*(X_1,X_2) | X_2) = 0
>     \end{equation}
>
> If the additive components $f^*_1 + f^*_2$ are small, say, in the extreme case $f^*_1 + f^*_2$ is zero, then $f^* = h^*$, and by the inequality above we know that the SID condition cannot be satisfied. Therefore, for the SID condition to be satisfied, it is necessary that the function $f^*$ has a significant amount of signal that is additive.
>
>  But of course, this is for the SID condition to be satisfied. It is possible that CART can still work well without the SID condition,
>     which needs more exploration and is beyond the scope of this work.
>
>
> Reply to questions:
>
> Thanks! We will correct them in the revision.

---

> > ### Comment · Reviewer_1fUr · 2023-08-15
> >
> > Thank you for your answer and your clarifications. I will keep my score unchanged.

---

### Decision · Program_Chairs · 2023-09-21

**Decision:**

Accept (poster)

**Comment:**

This paper ends up with all bordeline+ reviews. I believe the paper could have been improved because (i) the additive structure has indeed been made previously but stays as a limitation of the approach (if the authors had managed to weaken it, it would have undeniably been a +); (ii) focusing only on CART is a formidable restriction in the context of the paper: we know what are the properties of proper losses (for class probability estimation), of which CART’s is just an instantiation. Being able to generalise to proper losses would carry the tremendous potential to balance boosting rates and conditions for consistency.

I hope that upon acceptance, the authors will dig in that last question for a longer version of the paper: the path forward on covering not just CART's loss could yield to very interesting conclusions (the authors may keep in mind that a loss function gives lots of indications on learning with it, not just on consistency, but also training rates, stability of optimization, etc.).